# Four out of ten married women utilized modern contraceptive method in Ethiopia: A Multilevel analysis of the 2019 Ethiopia mini demographic and health survey

**Sewunet Sako Shagaro**[1]*, **Teshale Fikadu Gebabo**[2], **Be'emnet Tekabe Mulugeta**[1]

**1** Department of Health Informatics, School of Public Health, College of Medicine and Health Sciences, Arba Minch University, Arba Minch, Ethiopia, **2** Department of Public Health, School of Public Health, College of Medicine and Health Sciences, Arba Minch University, Arba Minch, Ethiopia

* zesew1lalem@gmail.com

**Data Availability Statement:** The survey dataset used in this analysis are third party data from the demographic and health survey website (https://

## Abstract

### Background

Modern contraceptive method is a product or medical procedure that interferes with reproduction from acts of sexual intercourse. Globally in 2019, 44% of women of reproductive age were using a modern method of contraception but it was 29% in sub-Saharan Africa. Therefore, the main aim of this analysis was to assess the prevalence of modern contraceptive utilization and associated factors among married women in Ethiopia.

### Method

The current study used the 2019 Ethiopia mini demographic and health survey dataset. Both descriptive and multilevel mixed-effect logistic regression analysis were done using STATA version 14. A p-value of less than 0.05 and an adjusted odds ratio with a 95% confidence interval were used to report statistically significant factors with modern contraceptive utilization.

### Result

The overall modern contraceptive utilization among married women in Ethiopia was 38.7% (95% CI: 37.3% to 40.0%). Among the modern contraceptive methods, injectables were the most widely utilized modern contraceptive method (22.82%) followed by implants (9.65%) and pills (2.71%). Maternal age, educational level, wealth index, number of living children, number of births in the last three years, number of under 5 children in the household, religion, and geographic region were independent predictors of modern contraceptive utilization.

### Conclusion

In the current study only four out of ten married non-pregnant women of reproductive age utilized modern contraceptive methods. Furthermore, the study has identified both individual

dhsprogram.com) and permission to access the data can be granted after sending a request for registration. Any interested researcher can obtain the data after completing the registration process (https://dhsprogram.com/data/dataset_admin/login_main.cfm) and submitting a short abstract of their proposed manuscript. The authors had no special access privileges to the data.

**Funding:** The authors received no specific funding for this work.

**Competing interests:** The authors have declared that no competing interests exist.

**Abbreviations:** AOR, Adjusted Odds Ration; CI, Confidence Interval; CSA, Central Statistical Agency; EMDHS, Ethiopia Mini Demographic and Health Survey; HSTP, Health Sector Transformation Plan; IUD, Interuterine Contraceptive Device; SDM, Standard Days Method.

and community-level factors that can affect the utilization of modern contraceptive methods by married women in the country. Therefore, concerned bodies need to improve access to reproductive health services, empower women through community-based approaches, and minimize region wise discrepancy to optimize the utilization.

## Introduction

Family planning is defined as the choice of individuals or couples to anticipate and attain their desired number of children, and the spacing and timing of their births. This can be realized through use of contraceptive methods and the treatment of involuntary infertility [1]. Ensuring universal access to sexual and reproductive health-care services, including for family planning, information and education, and the integration of reproductive health into national strategies and programmes; and reducing the maternal mortality ratio to less than 70 per 100,000 live births by 2030 are some of goals directly targeted to family planning mentioned under sustainable development goal (SDG) 3 of United Nations (UN) [2].

Modern contraceptive method is a product or medical procedure that interferes with reproduction from acts of sexual intercourse [3]. Modern methods of contraceptive include oral contraceptives, sterilization (male and female), interuterine contraceptive device (IUD), injectables, implants, condoms (male and female), diaphragm, lactational amenorrhea method, standard days method (SDM), emergency contraception, cervical caps, contraceptive sponges and other country specific modern methods [4].

Literatures have shown that contraceptive utilization can save the loss of more than 3 million girls per year due to unsafe abortion. It also improves school dropout, poverty, high rate of teenage marriage and complications related to unwanted pregnancy. It is believed that access to family planning methods gives women more freedom, independence and gender equity. Moreover, contraceptives utilization plays a significant role in improving maternal and child health, women empowerment, educational advances and economic development of the country. It allows the couples to decide on the desired number of children and to space births according to their plan and income level [5–8]. Therefore, realizing the rights and well-being of women can be assured by improving access to high quality, affordable sexual and reproductive health services and information mainly about the full range of family planning methods [9].

As indicated in the World Fertility and Family Planning 2020, globally 44% of women of reproductive age were utilizing modern contraceptive methods in 2019. The use of contraception among women of child bearing age in Sub-Saharan Africa increased from 13% in 1990 to 29% in 2019 [10]. The report also stated that the decrease of fertility in Sub-Saharan Africa has been quite slow as compared to other regions of the world. Consequently, Sub-Saharan Africa requires 34 years (from 1995 to 2029) to decline its fertility from 6.0 to 4.0 live births per woman while it took 24 years in Eastern and South-Eastern Asia for the same reduction. Also studies conducted in various countries of Africa revealed the prevalence of modern contraceptive methods utilization among married women continues to be low [11–15].

However, there was progress in utilization of contraceptives among women of reproductive age in the East Africa region on average from 21% (1990–94) to 39% (2010–14). The highest proportion of contraceptive users was reported from Kenya (53%) while the lowest users were from Uganda by 23% from the region [16, 17]. In east and other parts of Africa, injectables were the most widely utilized modern contraceptive method followed by implants [16, 18–21].

Various studies have identified maternal age, educational status, wealth index, births in the last three years, marital status, religion, region, and place of residence as individual-level and community-level factors associated with modern contraceptive utilization in Ethiopia and Africa as well [11, 15, 22, 23]. Other studies also considered husband's education level, media exposure, number of desired children and fertility related decisions as factors that may influence contraceptive utilization [13, 21, 24, 25]. As stated in literature, higher educational status of women was positively associated with modern contraceptive utilization where the odds of modern contraceptive utilization among women who attained primary educational levels was higher than women without formal education [12–15, 17, 26, 27]. Similarly, modern contraceptive utilization was highly influenced by the wealth index of the women [11, 14, 15, 25].

Despite the reduction of the total fertility rate over the years, Ethiopia has still one of the highest fertility rates in Africa. The prevalence of any modern contraceptive use by currently married Ethiopian women has tripled over the last fifteen years, jumping from 14% in 2005 to 41% in 2019 [28]. The most commonly used contraceptive methods for currently married women in the country are injectables (27%), followed by implants (9%). The lowest contraceptive utilization was reported from the Somali region (3%) and the highest from both Amhara and Oromia regions (50%) of Ethiopia [28]. However, as stated in the 2020 report of the Federal Ministry of Health of Ethiopia, the health sector transformation plan (HSTP) I target (55%) of the national contraceptive prevalence rate was not achieved. And all the regions in the country were unable to attain their target set for the year [29]. These evidence clearly convey that the modern contraceptive method utilization remains a great public health problem in Ethiopia. Therefore, this study aimed to assess the prevalence and associated factors of modern contraceptive utilization among married women in Ethiopia using the 2019 Ethiopia mini demographic and health survey data.

## Methods

### Study settings and data source

The study used the 2019 Ethiopia mini demographic and health survey (EMDHS) dataset. It is the second mini demographic and health survey which was conducted in March, 2019, to June, 2019 in Ethiopia. Ethiopia is divided into two administrative cities and nine regions. All married women in the reproductive age living in nine regions and two administrative cities of Ethiopia were included in the study.

### Sampling procedures

The EMDHS used a complete list of 149,093 enumeration areas (EAs) created for the upcoming Ethiopia population and housing census as a sampling frame. The frame comprises information about the EA location, type of residence (urban or rural), and estimated number of residential households. Accordingly, the sample was stratified and selected in two-stages [28].

Out of the total EAs, a total of 305 EAs (93 in urban areas and 212 in rural areas) were selected with probability proportional to EA size and with independent selection in each sampling stratum for the survey. Secondly, a fixed number of 30 households per cluster were selected with an equal probability systematic selection from the newly created household listing.

Finally, the 2019 EMDHS survey covered 8,663 households out of the selected 8,794 households providing a response rate of 99%. About 8,885 women completed the interview from 16,583 women identified for the interview, yielding a response rate of 99%. According to the EMDHS report response rates were higher in rural than in urban areas [28].

The source population of this study was all married non-pregnant women who were in the reproductive age group and living in Ethiopia. Pregnant mothers and those who were not in union at the time of survey were excluded from the study. Hence, 4,983 married women data were extracted from the 2019 EMDHS datasets.

## Measurements of variables

**Dependent variable.** Modern contraceptive utilization was the outcome variable of the study. Woman was considered as a "utilizer" if she had been utilizing any modern contraceptive methods such as oral contraceptives, male and female sterilization, intrauterine contraceptive device, injectables, implants, male and female condoms, lactational amenorrhea method, standard days method, and emergency contraception [4, 30] during the 2019 EMDHS survey period while woman who had been utilizing traditional, folkloric or no method was considered as a "non-utilizer".

**Independent variable.** Both the individual and community level explanatory variables were used to assess modern contraceptives utilized among women in childbearing age in the country.

**Individual level factors.** Maternal age, educational level of mother, wealth index, total children ever born, number of living children, births in the last three years, age of respondent at first birth, family size, number of under five children in household, and knowledge on modern contraceptive method.

**Community-level variables.** Community level explanatory variables include religion, region, and place of residence (Table 1).

## Data processing and statistical analysis

The data were extracted, cleaned, re-coded, and analyzed using STATA version 14. Descriptive statistics were presented using graphs, tables and narrations. A multilevel mixed-effect logistic regression analysis, an advanced model, was used to overcome the violation of independence of observations and equal variance assumption of the traditional logistic regression model due to a hierarchical nature of DHS data.

We first estimated an intercept-only model or the null model (with only the outcome variable). Secondly, we included all individual level factors in the model (model I). In the third stage, we constructed model II (fitted with community-level factors only) and finally model III was fitted with both individual and community-level factors. To examine clustering and the extent to which community-level factors explain the unexplained variance of the null model, the intraclass correlation coefficient (ICC) and a proportional change in variance (PCV) were checked. The model with the lowest deviance, model III, was selected as the best-fitted model for the analysis. Variance inflation factor (VIF) was done to test the existence of multicollinearity among the independent variables.

Variables having a p-value of less than 0.2 in the bivariable analysis were selected as candidate variables for the multivariable mixed-effect logistic regression analysis. In the final model, a p-value of less than 0.05 and an adjusted odds ratio (AOR) with a 95% confidence interval (CI) were used to report statistically significant factors with modern contraceptive utilization among childbearing age women.

## Ethical consideration

This study used secondary data from demographic and health survey repositories. The survey data do not contain all identifying information. The MEASURE Demographic and Health

**Table 1. Description of individual and community level variables.**

| Variables | Description |
|---|---|
| **Individual-level variables** | |
| Maternal age | It is the current age of women recoded as 15–19, 20–24, 25–29, 30–34, 35–39, 40–44, and 45–49. |
| Educational level of mother | This is the level of education a woman attained and recoded as no education, primary, secondary and higher. |
| Wealth index | In DHS the wealth index is calculated using data on a household's ownership of selected assets. Each household asset is assigned a weight score generated through PCA. The resulting asset scores are standardized and summed by household and individuals are ranked according to the total score of the household in which they reside. Finally, it is grouped as poorest, poorer, middle, richer and richest. It is recoded as Low Income (poorest and poorer), Middle Income and High Income (richer and richest). |
| Total children ever born | It is recoded as 1–3 children, 4–6 children, and 7 and more |
| Number of living children | Number of living children is recoded as no child, 1–4, 5–8, 9 and above children. |
| Births in the last three years | This variable is recoded as no birth, one birth, and 2 and above births |
| Age of respondent at 1st birth | Recoded as less than 18 years, 18–24, and 25 and above years |
| Family size | It is number of household members and recoded as 1–4, 5–8, and 9 and above |
| Number of <5 children in household | Number of children 5 and under in household is recoded as none, 1–2, and 3 and above children. |
| Knowledge on modern contraceptive method | Knowledge of contraceptive method is defined as percentage who know of any method, any modern method, any traditional method and specific methods; mean number of methods known. It is recoded as "yes" if the women say they know or have heard at least one of the modern contraceptive methods and else recoded as "no". |
| Religion | This variable is the religious group to which the woman associates herself and recoded as Muslim, Orthodox, Protestant, Catholic and other. |
| Region | Region of residence is typically the first administrative level within the country, or a grouping of the first administrative level. It is grouped as Tigray, Afar, Amhara, Oromia, Somali, Benishangul, SNNPR, Gambela, Harari, Addis Ababa and Dire Dawa. |
| Place of residence | It is the designation of the cluster or enumeration area as an urban area or a rural area. |

PCA: principal components analysis.

Survey Program team allowed us to access the data upon sending an abstract of our study to an online request form http://www.measuredhsprogram.com.

# Results

## Socio-demographic characteristics of the study participants

A total of 4,983 non-pregnant married women were included in the analysis. The mean age of respondents was 30.7 years (SD ± 8.29 years). From the total number of respondents, almost one quarter (23.48%) were in the age range of 25–29 years. More than half (52.26%) of the women included in the analysis had no education whereas only 314(6.30%) respondents attended higher education. In this study, women who were from low income households (41.26%) were nearly equal to women who were from high income households (42.83%).

More than one quarter (26.73%) of the respondents had 1–3 children ever born and about 2965 (59.5%) of the women have had 1–4 number of live children. One half of the respondents (49.75%) had no birth while 352 (7.06%) respondents had two and more births in the last three years. Out of the total respondents, one half (50.35%) were in the age range of less than 18

years whereas 842(16.9%) were in the age range of 25 and above years during their first birth. Regarding family size, 2,652(53.22%) households have had 5–8 family members and 3,088 (61.97%) households have had 1 or 2 child/children aged less than 5 years. Majority of the women (93.62%) included in the survey know modern contraceptives methods.

Concerning the community-level factors, the majority (42.24%) were followers of Muslim religion and nearly three fourth (72.61%) of the women were rural residents. Slightly higher proportion of women (12.88%) were from Oromia region and the lowest proportion (6.64%) were from Addis Ababa city administration (Table 2).

**Table 2. Distribution of individual and community-level factors of the married women in Ethiopia, EMDHS 2019.**

| Individual variables | Number (N) | Percent (%) |
|---|---|---|
| **Maternal age** | | |
| 15–19 | 382 | 7.67 |
| 20–24 | 803 | 16.11 |
| 25–29 | 1170 | 23.48 |
| 30–34 | 870 | 17.46 |
| 35–39 | 810 | 16.26 |
| 40–44 | 546 | 10.96 |
| 45–49 | 402 | 8.07 |
| **Educational level of mother** | | |
| No education | 2604 | 52.26 |
| Primary | 1589 | 31.89 |
| Secondary | 476 | 9.55 |
| Higher | 314 | 6.30 |
| **Wealth index** | | |
| Low Income | 2056 | 41.26 |
| Middle Income | 793 | 15.91 |
| High Income | 2134 | 42.83 |
| **Total children ever born** | | |
| 1–3 children | 2189 | 43.93 |
| 4–6 children | 1462 | 29.34 |
| 7 and above | 1332 | 26.73 |
| **Number of living children** | | |
| No child | 514 | 10.32 |
| 1–4 children | 2965 | 59.50 |
| 5–8 children | 1391 | 27.91 |
| 9 and above | 113 | 2.27 |
| **Births in the last three years** | | |
| No birth | 2479 | 49.75 |
| One birth | 2152 | 43.19 |
| 2 and above births | 352 | 7.06 |
| **Age of respondent at first birth** | | |
| Less than 18 years | 2509 | 50.35 |
| 18–24 | 1632 | 32.75 |
| 25 and above | 842 | 16.90 |
| **Number of household members** | | |
| 1–4 | 1803 | 36.18 |
| 5–8 | 2652 | 53.22 |
| 9 and above | 528 | 10.60 |

*(Continued)*

**Table 2.** (Continued)

| Individual variables | Number (N) | Percent (%) |
|---|---|---|
| **Number of under five children in household** | | |
| None | 1521 | 30.52 |
| 1–2 | 3088 | 61.97 |
| 3 and above | 374 | 7.51 |
| **Knowledge of any method** | | |
| Knows no method | 311 | 6.24 |
| Knows only folkloric method | 1 | 0.02 |
| Knows only traditional method | 6 | 0.12 |
| Knows modern method | 4665 | 93.62 |
| **Religion** | | |
| Muslim | 2,105 | 42.24 |
| Orthodox | 1,806 | 36.24 |
| Protestant | 983 | 19.73 |
| Catholic | 38 | 0.76 |
| Other | 51 | 1.02 |
| **Region** | | |
| Tigray | 391 | 7.85 |
| Afar | 407 | 8.17 |
| Amhara | 584 | 11.72 |
| Oromia | 642 | 12.88 |
| Somali | 353 | 7.08 |
| Benishangul | 467 | 9.37 |
| SNNPR | 629 | 12.62 |
| Gambela | 406 | 8.15 |
| Harari | 378 | 7.59 |
| Addis Ababa | 331 | 6.64 |
| Dire Dawa | 395 | 7.93 |
| **Type of place of residence** | | |
| Urban | 1365 | 27.39 |
| Rural | 3618 | 72.61 |

## Prevalence of modern contraceptive utilization

From the total women included in the analysis, 1,927 (38.7%) respondents utilized modern contraceptive methods. In this study, non-modern contraceptive users were those who utilized traditional methods, folkloric methods or no method at the time of the survey. Accordingly, a large number of women did not utilize any contraceptive method while the least commonly utilized method was folkloric method (Fig 1). Furthermore, injectables were the most widely utilized modern contraceptive method (22.82%) followed by implants (9.65%), and pills (2.71%). (Fig 2).

## Factors affecting modern contraceptive utilization

From all independent individual and community level factors, maternal age, educational level of women, wealth index, total children ever born, births in the last three years, age of mother at first birth, number of household members, number of under five children in household, religion, region and place of residence were eligible variables for multivariable multilevel analysis.

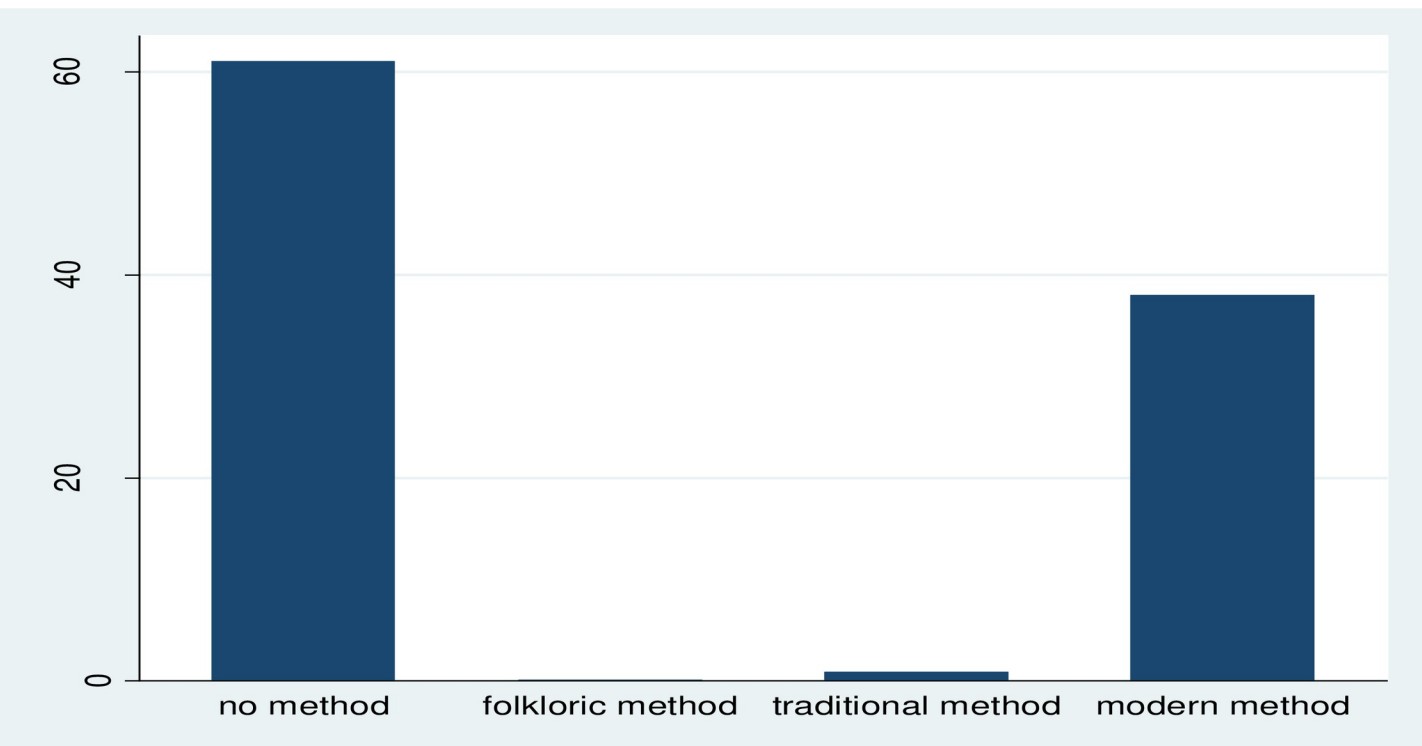

**Fig 1. Percentage of current use by contraceptive method type among married women in Ethiopia, EMDHS 2019.**

In the multivariable multilevel analysis, the following individual-level and community-level factors were associated with modern contraceptive utilization in Ethiopia. The odds of modern contraceptive utilization was 47% [AOR = 0.53; 95%CI: 0.36–0.79], 67% [AOR = 0.33; 95%CI: 0.21–0.51], and 87% [AOR = 0.13; 95%CI: 0.08–0.22] lower among women who aged 35–39, 40–44, and 45–49 years as compared to women aged 15–19 years. A woman who attended primary, secondary, and higher school was 1.73 [AOR = 1.73; 95%CI: 1.45–2.06], 1.77 [AOR = 1.77; 95%CI: 1.34–2.33] and 2.56 [AOR = 2.56; 95%CI: 1.83–3.59] times more likely to utilize modern contraceptive method as compared to woman who had not attended formal education. The odds of modern contraceptive utilization among women from households with middle and high income status were 1.72 [AOR = 1.72; 95%CI: 1.38–2.15] and 1.88 [AOR = 1.88; 95%CI: 1.51–2.34] times higher as compared to those from households with low income status. Those women who had 1–4, 5–8, and 9 and above living children were 2.51 [AOR = 2.51; 95%CI: 1.48–4.25], 2.67 [AOR = 2.67; 95%CI: 1.56–4.56] and 4.13 [AOR = 4.13; 95%CI: 1.94–8.80] times more likely utilize modern contraceptive method as compared to those women who had no living child. Woman who had one birth, and two and above births in the last three years was 29% [AOR = 0.71; 95%CI: 0.59–0.86] and 47% [AOR = 0.53; 95%CI: 0.37–0.75] lowers odds of modern contraceptive utilization as compared to those woman who had no birth in the last three years. The odds of modern contraceptive utilization was 1.8 [AOR = 1.80; 95%CI: 1.46–2.22] times higher among women from households where the number of under five children were 1–2 as compared to women from households where there were no under five children.

Among community level factors, the odds of modern contraceptive method utilization was 1.86 [AOR = 1.86; 95%CI: 1.46–2.36] and 1.37 [AOR = 1.37; 95%CI: 1.02–1.83] times higher among women who have been following Orthodox religion and Protestant religion as

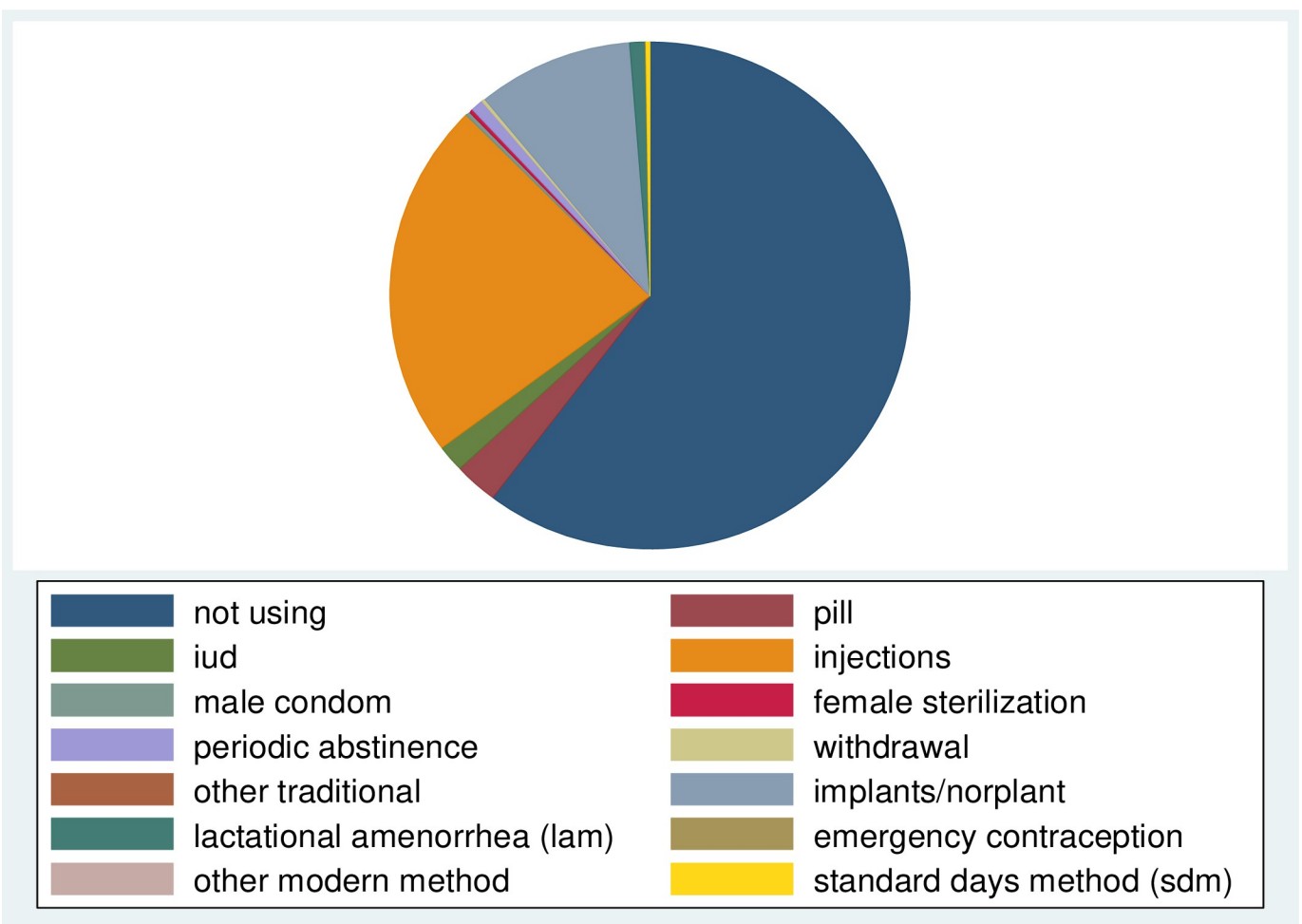

**Fig 2. Proportion of current use by contraceptive method among married women in Ethiopia, EMDHS 2019.**

compared to women who have been following Muslim religion. The odds of modern contraceptive method utilization among women living in Amhara, Oromia, Benishangul and Southern nation nationalities and peoples regions of Ethiopia were 2.33 [AOR = 2.33; 95%CI: 1.43–3.80], 2.06 [AOR = 2.06; 95%CI: 1.24–3.43], 1.86 [AOR = 1.86; 95%CI: 1.09–3.18], and 2.35 [AOR = 2.35; 95%CI: 1.40–3.94] times higher as compared to women living in Tigray region, respectively. The odds of modern contraceptive method utilization among women living in Afar and Somali regions were lower by 59% [AOR = 0.41; 95%CI: 0.22–0.78] and 88% [AOR = 0.12; 95%CI: 0.05–0.27] as compared to women living in Tigray region of Ethiopia (Table 3).

Regarding model fitness, as depicted in Table 4, intra-cluster correlation coefficient (ICC) of 28.97% (in the null model) confirmed the choice of multilevel model for the analysis. It indicates that 28.97% of the variation in modern contraceptive utilization among married women was due to the variation between clusters. Also the highest PCV (93.70%) in the last model (Model III) show most of the variations of modern contraceptive utilization were attributable to both individual and community level factors. Furthermore, the lowest deviance in the final model confirmed that the model (Model III) was the best fitted-model (Table 4).

**Table 3. Multivariable multilevel analysis for factors associated with modern contraceptive utilization among married women in Ethiopia, EMDHS 2019.**

| Variables | Null Model | Model I AOR (95% CI) | Model II AOR (95%CI) | Model III AOR (95%CI) |
|---|---|---|---|---|
| **Maternal age** | | | | |
| 15–19 | | 1.00 | | 1.00 |
| 20–24 | | 1.27 [0.92–1.74] | | 1.23 [0.89–1.69] |
| 25–29 | | 1.01 [0.73–1.41] | | 0.92 [0.67–1.28] |
| 30–34 | | 0.86 [0.60–1.24] | | 0.77 [0.54–1.11] |
| 35–39 | | 0.61 [0.41–0.91] | | 0.53 [0.36–0.79]** |
| 40–44 | | 0.39 [0.25–0.60] | | 0.33 [0.21–0.51]*** |
| 45–49 | | 0.15 [0.09–0.25] | | 0.13 [0.08–0.22]*** |
| **Educational level of mother** | | | | |
| No education | | 1.00 | | 1.00 |
| Primary | | 1.73 [1.45–2.06] | | 1.62 [1.35–1.93] *** |
| Secondary | | 1.77 [1.34–2.33] | | 1.50 [1.14–1.98]** |
| Higher | | 2.56 [1.83–3.59] | | 2.06 [1.47–2.88] *** |
| **Wealth index** | | | | |
| Low Income | | 1.00 | | 1.00 |
| Middle Income | | 1.72 [1.38–2.15] | | 1.50 [1.21–1.86]*** |
| High Income | | 1.88 [1.51–2.34] | | 1.63 [1.30–2.05]*** |
| **Total children ever born** | | | | |
| 1–3 children | | 1.00 | | 1.00 |
| 4–6 children | | 1.13 [0.90–1.42] | | 1.14 [0.90–1.43] |
| 7 and above | | 0.99 [0.70–1.39] | | 1.01 [0.72–1.42] |
| **Number of living children** | | | | |
| No child | | 1.00 | | 1.00 |
| 1–4 children | | 2.57 [1.52–4.35] | | 2.51 [1.48–4.25]** |
| 5–8 children | | 2.69 [1.58–4.58] | | 2.67 [1.56–4.56]*** |
| 9 & above | | 4.18 [1.97–8.90] | | 4.13 [1.94–8.80]*** |
| **Births in the last three years** | | | | |
| No birth | | 1.00 | | 1.00 |
| One birth | | 0.72 [0.60–0.87] | | 0.71 [0.59–0.86] *** |
| 2 and above births | | 0.47 [0.33–0.67] | | 0.53 [0.37–0.75] *** |
| **Age of respondent at first birth** | | | | |
| Less than 18 years | | 1.00 | | 1.00 |
| 18–24 | | 1.24 [1.04–1.47] | | 1.28 [1.08–1.52] |
| 25 and above | | 0.96 [0.70–1.30] | | 0.93 [0.68–1.26] |
| **Number of household members** | | | | |
| 1–4 | | 1.00 | | 1.00 |
| 5–8 | | 0.95 [0.79–1.14] | | 0.94 [0.79–1.13] |
| 9 and above | | 0.86 [0.63–1.18] | | 0.90 [0.66–1.24] |
| **Number of under five children in household** | | | | |
| None | | 1.00 | | 1.00 |
| 1–2 | | 1.79 [1.45–2.22] | | 1.80 [1.46–2.22] *** |
| 3 and above | | 0.90 [0.60–1.36] | | 1.02 [0.68–1.54] |
| **Religion** | | | | |
| Muslim | | | 1.00 | 1.00 |
| Orthodox | | | 1.75 [1.40–2.20] | 1.86 [1.46–2.36] *** |
| Protestant | | | 1.41 [1.07–1.85] | 1.37 [1.02–1.83] |
| Catholic | | | 1.80 [0.81–4.01] | 2.12 [0.91–4.92] |

*(Continued)*

**Table 3.** (Continued)

| Variables | Null Model | Model I AOR (95% CI) | Model II AOR (95%CI) | Model III AOR (95%CI) |
|---|---|---|---|---|
| Other | | | 0.76 [0.36–1.59] | 0.92 [0.42–1.98] |
| **Region** | | | | |
| Tigray | | | 1.00 | 1.00 |
| Afar | | | 0.32 [0.17–0.59] | 0.41 [0.22–0.78]** |
| Amhara | | | 1.82 [1.13–2.93] | 2.33 [1.43–3.80]** |
| Oromia | | | 1.73 [1.05–2.84] | 2.06 [1.24–3.43]** |
| Somali | | | 0.07 [0.03–0.16] | 0.12 [0.05–0.27] *** |
| Benishangul | | | 1.57 [0.93–2.66] | 1.86 [1.09–3.18]* |
| SNNPR | | | 2.02 [1.21–3.35] | 2.35 [1.40–3.94] ** |
| Gambela | | | 1.10 [0.63–1.90] | 1.07 [0.61–1.88] |
| Harari | | | 1.11 [0.64–1.95] | 1.11 [0.62–1.96] |
| Addis Ababa | | | 1.48 [0.83–2.66] | 1.68 [0.93–3.06] |
| Dire Dawa | | | 0.87 [0.49–1.53] | 1.06 [0.59–1.88] |
| **Type of place of residence** | | | | |
| Urban | | | 1.00 | 1.00 |
| Rural | | | 0.77 [0.40–0.72] | 0.77 [0.55–1.06] |

*P<0.05

**P<0.001

***P<0.0001.

## Discussion

The prevalence of modern contraceptive utilization among married women in Ethiopia was 38.7% (95% CI: 37.3% to 40.0%). This finding is higher than the result of the secondary data analysis of the 2016 Ethiopia demographic and health survey where the prevalence of modern contraceptive utilization was 20.42% [11]. It was also higher than the studies conducted in different parts of Ethiopia like Bale zone (20.8%), Metekel zone (18.6%), and Afar region (8.5%); and other African countries such as Ghana (21%), Burkina Faso (24%), Senegal (26.3%) and Nigeria (10.3%) [13–15, 22, 26, 31, 32]. The current prevalence was higher than the national figure (35%) according to the 2016 EDHS report and the findings of the secondary data analysis of the 2016 EDHS (34.9%) [23, 30]. However, it was similar to a study done in Northwest Ethiopia, which was 37% [33]. On the other hand, the utilization rate was lower when compared to the study conducted in Western Ethiopia (71.9%) [24], Southern Ethiopia (69.5%,

**Table 4. Random effect analysis and model fitness for assessing factors associated with modern contraceptive utilization among married women in Ethiopia, EMDHS 2019.**

| Parameters | Null Model | Model I | Model II | Model III |
|---|---|---|---|---|
| Community level variance (SE) | 0.16 | 0.1458 | 0.0819 | 0.0864 |
| ICC | 0.2897 | 0.2453 | 0.1441 | 0.1411 |
| PCV | Reference | 54.89% | 92.95% | 93.70 |
| **Model fitness** | | | | |
| Log-likelihood | –3039.41 | –2775.74 | –2927.37 | –2683.76 |
| Deviance | 6078.82 | 5551.48 | 5854.74 | 5367.52 |
| AIC | 6082.82 | 5603.48 | 5888.75 | 5449.52 |
| BIC | 6095.85 | 5772.85 | 5999.48 | 5716.58 |

53.3%) [19, 34], Northern Ethiopia (51.3%, 66.2%) [18, 25], Yaoundé-Cameroon (58.9%) [35], and India (48.6%) [36]. These differences could be due to participants' cultural, religious and awareness differences on modern contraceptive methods. Additionally, variation of the study period might contributed for the observed discrepancy of modern contraceptive methods utilization.

As evidenced by the literature, the prevalence of unplanned pregnancy is lower among younger women than older women [37]. This fact is supported by our study finding where the odds of modern contraceptive utilization was lower among aged women as compared to young women. This finding is also in agreement with the study findings done in Amhara regional state of Ethiopia [25], Gondar [38], East Africa [17] and findings from secondary data analysis of the 2016 EDHS [23]. The possible reason might be related with the woman's educational status. In our study, aged women were less educated as compared to the young women. Previous studies have reported that as maternal educational level increase the likelihood of desire for bearing more children decreases [39]. Also there are evidences that show aged women may be in less need of modern contraceptive methods as they prefer traditional methods than the young women [40].

Our finding also revealed that the educational level of the mother was an independent predictor of modern contraceptive method utilization among married women in Ethiopia. Women who attended primary, secondary and higher education were more likely to utilize modern contraceptive methods as compared to women with no education. This finding is in line with studies conducted in Western Ethiopia [24], Northwest Ethiopia [26, 32], Ghana [12, 41] and Nigeria [14]. This might be due to an increase in the level of education, enhances awareness on sexual and reproductive health, maximizes probability of getting contraceptive related information and improves decision making ability of women.

In this study, household wealth had effects on modern contraceptive method utilization where the odds of modern contraceptive utilization among married women from households with middle and high income status were higher than women from households with low income status. This finding was concordant with the findings from secondary data analysis of Ethiopia demographic and health survey [25, 30], three demographic and health surveys in Bangladesh [42], a community based cross-sectional study done in Western Ethiopia [24], Eastern Ethiopia [32] and Nigeria [14]. In Ethiopia, all public health facilities provide modern contraceptives for any woman in the child bearing age (15–49 years) free of charge, hence household income status has no effect in obtaining the service. However, a woman household's wealth has great influence on exposure to education, access to basic health services and health information.

In this study, the odds of modern contraceptive utilization among women who had one or more children were higher as compared to women who had no child. This finding is supported by the study finding conducted in Southern nation nationalities and peoples region [19], North Shoa zone [43] and Senegal [15]. This could be due to women who had one or more children might have greater need to space or limit births than those women who had no child. This evidence is also supported by the current study which reported that a woman from a household with one or two under five children was more likely to utilize modern contraceptive method as compared to a woman who had no child aged less than five.

This study revealed that births in the last three years were significantly associated with modern contraceptive utilization. Women who had one birth or who had two and more births were 33% and 47% times less likely to utilize modern contraceptive methods as compared to women who had no birth in the last three years respectively. The finding is consistent with the result of the study conducted in Ethiopia where modern contraceptive utilization was lower among women who had one birth or two and more births as compared to women who had no

birth in the last three years [23, 43]. The possible reason for this could be the variation in their desire to have more children. The problem of reverse causality might be also another reason as most of the women who reported as present utilizers may started utilizing the methods two or three years ago.

Multivariate multilevel analysis also showed that religion of the woman was independently associated with modern contraceptive utilization. Women who had been following Orthodox religion and Protestant religion had higher odds of utilizing modern contraceptives as compared to those women who had been following Muslim religion. This finding was consistent with secondary data analysis of three demographic and health survey in Bangladesh where Muslim women have lower odds of contraceptive utilization compared with non-Muslim women [42, 44]. This finding was also in agreement with secondary data analysis of Ethiopia demographic and health survey in 2016 [23].

Women's region of residence was also associated with modern contraceptive method utilization among married women in Ethiopia. Women who live in Afar and Somali regions of Ethiopia were less likely to utilize modern contraceptive methods as compared to women who live in Tigray region. In contrast, women who reside in Amhara, Oromia, Benishangul and SNNPR regions of Ethiopia were more likely to utilize modern contraceptive methods as compared to women who live in Tigray region. The finding was in line with secondary data analysis of the 2016 Ethiopia demographic and health survey where a woman who resides in the Afar and Somali regions was nearly 55% and 93% times less likely to utilize modern contraceptive as compared to a woman who resides in Addis Ababa [11].

The strength of the current study includes use of multilevel mixed effect analysis to overcome the hierarchical nature of EMDHS data, and deploying nationally weighted representative and most recent EMDHS data which shows country level proportion of contraceptive utilization and its associated factors among women in child bearing age. The study also has some limitations. Showing temporal relationship between modern contraceptives utilization and its predictors was impossible due to the type of the study design, cross-sectional, used for the survey. Moreover, the EMDHS data did not encompass information about some predictors of modern contraceptive utilization, as it was a mini report. Conversely, the investigators have trust that the stated limitations cannot impose significant impact on the validity of study findings.

## Conclusion

In the current study only four out of ten married non-pregnant women of reproductive age utilized modern contraceptive methods. Furthermore, the study has identified both individual and community-level factors that can affect the utilization of modern contraceptive methods by married women in the country. Therefore, concerned bodies need to improve access to reproductive health services, empower women through community-based approaches, and minimize region wise discrepancy to optimize the utilization.

## Acknowledgments

We would like to forward our kindest regards to MEASURE Demographic and Health Survey Program for providing us 2019 EMDHS data.

## Author Contributions

**Conceptualization:** Sewunet Sako Shagaro.

**Data curation:** Sewunet Sako Shagaro.

**Formal analysis:** Sewunet Sako Shagaro, Teshale Fikadu Gebabo, Be'emnet Tekabe Mulugeta.

**Investigation:** Sewunet Sako Shagaro, Teshale Fikadu Gebabo, Be'emnet Tekabe Mulugeta.

**Methodology:** Sewunet Sako Shagaro, Teshale Fikadu Gebabo, Be'emnet Tekabe Mulugeta.

**Project administration:** Sewunet Sako Shagaro.

**Software:** Sewunet Sako Shagaro, Teshale Fikadu Gebabo, Be'emnet Tekabe Mulugeta.

**Validation:** Sewunet Sako Shagaro.

**Writing – original draft:** Sewunet Sako Shagaro, Teshale Fikadu Gebabo, Be'emnet Tekabe Mulugeta.

**Writing – review & editing:** Sewunet Sako Shagaro, Teshale Fikadu Gebabo, Be'emnet Tekabe Mulugeta.

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
