## [Decision Letter · Decision Letter 0]

4 Oct 2021

PONE-D-21-20183"Four out of Ten Married Women Utilized Modern Contraceptive Method in Ethiopia: A Multilevel Analysis Using the Ethiopia Mini Demographic and Health Survey 2019".PLOS ONE

Dear Dr. Shagaro,

Thank you for submitting your manuscript to PLOS ONE. After careful consideration, we feel that it has merit but does not fully meet PLOS ONE’s publication criteria as it currently stands. Therefore, we invite you to submit a revised version of the manuscript that addresses the points raised during the review process.

Major Revisions

We look forward to receiving your revised manuscript.

Kind regards,

Faisal Abbas, PhD

Academic Editor

PLOS ONE

Additional Editor Comments (if provided):

Major Revisions required.

Journal Requirements:

2. PLOS ONE does not copy edit accepted manuscripts (https://journals.plos.org/plosone/s/criteria-for-publication#loc-5). To that effect, please ensure that your submission is free of typos and grammatical errors.

4. We note you have included a table to which you do not refer in the text of your manuscript. Please ensure that you refer to Table 1 in your text; if accepted, production will need this reference to link the reader to the Table.

Reviewers' comments:

Reviewer's Responses to Questions

**Comments to the Author**

1. Is the manuscript technically sound, and do the data support the conclusions?

Reviewer #1: Yes

Reviewer #2: Partly

2. Has the statistical analysis been performed appropriately and rigorously? 

Reviewer #1: I Don't Know

Reviewer #2: No

3. Have the authors made all data underlying the findings in their manuscript fully available?

Reviewer #1: Yes

Reviewer #2: Yes

4. Is the manuscript presented in an intelligible fashion and written in standard English?

Reviewer #1: No

Reviewer #2: No

5. Review Comments to the Author

Reviewer #1: This study aims to assess the prevalence and associated factors of modern contraceptive utilization among married women in Ethiopia using the 2019 Ethiopia Mini Demographic and Health Survey data. Findings identify a number of significant relationships between independent and outcome variables that conform to other studies in the region, while indicators on prevalence differ from other country studies.

The study approach merits the importance of utilizing secondary datasets for analyzing in more depth DHS data and particularly the application of multivariate analysis in identifying key variables for further investigation. Noted is the higher contraception rate identified in your analysis to that reported in other studies. Could you clarify if this was due to the time at which these studies were conducted or was the variance in findings across studies due to framing of questions of studies which identified lower levels lower levels of contraceptive use?

Conclusions are supported by other regional and international findings in the area but could be more fully examined in terms of their implications for future interventions to further build contraceptive use in Ethiopia.

Minor comments is that the manuscript would benefit from editing for grammar. Eg: “those women who were born children” Does it mean "those women who had borne children? “First wealth quantile” Does this mean highest wealth quintile? “Poorest and Richest” are loaded terms - perhaps use “Low Income” Low -to Middle Income”, ”Middle Income”, and “High Income” as per the income classifications in other national surveys.

Reviewer #2: Please thoroughly revise the manuscript for language and grammar correction. Use appropriate and more scientific words like 'reported' instead of 'declared'. In Introduction, line (32), use appropriate word like 'choice' instead of 'ability'. CLearly mention which methods are considered as 'Modern Contraceptive Methods'. In Results, please merge table 1 and 2 to avoid clutter, though they represent individual and community level results, dividing it into two sections with a heading using a single row with heading. Line 185: 'did not utilize'. Statement in lines 279-283 completely contradict itself.

6. PLOS authors have the option to publish the peer review history of their article (what does this mean?). If published, this will include your full peer review and any attached files.

Reviewer #1: **Yes: **Dr Tahir Turk Adjunct Professor Dow University of Health Sciences, School of Public Health

Reviewer #2: **Yes: **Muhammad Tahir Khan

---

## [Author Response · Author response to Decision Letter 0]

18 Oct 2021

Dear sir/madam,

Thanks for your very constructive and pertinent feedback on our manuscript. I have tried to modify and correct our manuscript as per the given comments. Here below, you can find my response to each of your concerns. Thanks a lot! 

Academic Editor:

Comment:

Response:

We have tried to modify the full contents of our manuscript according to PLOS ONE’s style requirements 

Four out of ten married women utilized modern contraceptive method in Ethiopia: A Multilevel analysis of the 2019 Ethiopia mini demographic and health survey 

Sewunet Sako Shagaro1*, Teshale Fikadu Gebabo2 and Be’emnet Tekabe Mulgeta1

1Department of Health Informatics, School of Public Health, College of Medicine and Health Sciences, Arba Minch University, Arba Minch, Ethiopia

2Department of Public Health, School of Public Health, College of Medicine and Health Sciences, Arba Minch University, Arba Minch, Ethiopia 

* zesew1lalem@gmail.com (SS) 

Abstract 

Background: Modern contraceptive method is a product or medical procedure that interferes with reproduction from acts of sexual intercourse. Globally in 2019, 44% of women of reproductive age were using a modern method of contraception but it was 29% in sub-Saharan Africa. Therefore, the main aim of this analysis was to assess the prevalence of modern contraceptive utilization and associated factors among married women in Ethiopia.

Method: The current study used the 2019 Ethiopia mini demographic and health survey dataset. Both descriptive and multilevel mixed-effect logistic regression analysis were done using STATA version 14. A p-value of less than 0.05 and an adjusted odds ratio with a 95% confidence interval were used to report statistically significant factors with modern contraceptive utilization.

Result: The overall modern contraceptive utilization among married women in Ethiopia was 38.7% (95% CI: 37.3% to 40.0%). Among the modern contraceptive methods, injectables were the most widely utilized modern contraceptive method (22.82%) followed by implants (9.65%) and pills (2.71%). Maternal age, educational level, wealth index, number of living children, number of births in the last three years, number of under 5 children in the household, religion, and geographic region were independent predictors of modern contraceptive utilization. 

Conclusion: In the current study only four out of ten married non-pregnant women of reproductive age utilized modern contraceptive methods. Furthermore, the study has identified both individual and community-level factors that can affect the utilization of modern contraceptive methods by married women in the country. Therefore, concerned bodies need to improve access to reproductive health services, empower women through community-based approaches, and minimize region wise discrepancy to optimize the utilization.

Keywords: modern contraceptive method, reproductive age, Ethiopia 

Introduction 

Family planning is defined as the choice of individuals or couples to anticipate and attain their desired number of children, and the spacing and timing of their births. This can be realized through use of contraceptive methods and the treatment of involuntary infertility(1). Ensuring universal access to sexual and reproductive health-care services, including for family planning, information and education, and the integration of reproductive health into national strategies and programmes; and reducing the maternal mortality ratio to less than 70 per 100,000 live births by 2030 are some of goals directly targeted to family planning mentioned under sustainable development goal (SDG) 3 of United Nations (UN)(2). 

Modern contraceptive method is a product or medical procedure that interferes with reproduction from acts of sexual intercourse(3). Modern methods of contraceptive include oral contraceptives, sterilization (male and female), interuterine contraceptive device (IUD), injectables, implants, condoms (male and female), diaphragm, lactational amenorrhea method, standard days method (SDM), emergency contraception, cervical caps, contraceptive sponges and other country specific modern methods(4). 

Literatures have shown that contraceptive utilization can save the loss of more than 3 million girls per year due to unsafe abortion. It also improves school dropout, poverty, high rate of teenage marriage and complications related to unwanted pregnancy. It is believed that access to family planning methods gives women more freedom, independence and gender equity. Moreover, contraceptives utilization plays a significant role in improving maternal and child health, women empowerment, educational advances and economic development of the country. It allows the couples to decide on the desired number of children and to space births according to their plan and income level(5–8). Therefore, realizing the rights and well-being of women can be assured by improving access to high quality, affordable sexual and reproductive health services and information mainly about the full range of family planning methods(9). 

As indicated in the World Fertility and Family Planning 2020, globally 44% of women of reproductive age were utilizing modern contraceptive methods in 2019. The use of contraception among women of child bearing age in Sub-Saharan Africa increased from 13% in 1990 to 29% in 2019(10). The report also stated that the decrease of fertility in Sub-Saharan Africa has been quite slow as compared to other regions of the world. Consequently, Sub-Saharan Africa requires 34 years (from 1995 to 2029) to decline its fertility from 6.0 to 4.0 live births per woman while it took 24 years in Eastern and South-Eastern Asia for the same reduction. Also studies conducted in various countries of Africa revealed the prevalence of modern contraceptive methods utilization among married women continues to be low(11–15). 

However, there was progress in utilization of contraceptives among women of reproductive age in the East Africa region on average from 21% (1990-94) to 39% (2010-14). The highest proportion of contraceptive users was reported from Kenya (53%) while the lowest users were from Uganda by 23% from the region(16,17). In east and other parts of Africa, injectables were the most widely utilized modern contraceptive method followed by implants(16,18–21). 

Various studies have identified maternal age, educational status, wealth index, births in the last three years, marital status, religion, region, and place of residence as individual-level and community-level factors associated with modern contraceptive utilization in Ethiopia and Africa as well(11,15,22,23). Other studies also considered husband’s education level, media exposure, number of desired children and fertility related decisions as factors that may influence contraceptive utilization(13,21,24,25). As stated in literature, higher educational status of women was positively associated with modern contraceptive utilization where the odds of modern contraceptive utilization among women who attained primary educational levels was higher than women without formal education(12–15,17,26,27). Similarly, modern contraceptive utilization was highly influenced by the wealth index of the women(11,14,15,25). 

Despite the reduction of the total fertility rate over the years, Ethiopia has still one of the highest fertility rates in Africa. The prevalence of any modern contraceptive use by currently married Ethiopian women has tripled over the last fifteen years, jumping from 14% in 2005 to 41% in 2019(28). The most commonly used contraceptive methods for currently married women in the country are injectables (27%), followed by implants (9%). The lowest contraceptive utilization was reported from the Somali region (3%) and the highest from both Amhara and Oromia regions (50%) of Ethiopia(28). However, as stated in the 2020 report of the Federal Ministry of Health of Ethiopia, the health sector transformation plan (HSTP) I target (55%) of the national contraceptive prevalence rate was not achieved. And all the regions in the country were unable to attain their target set for the year(29). These evidence clearly convey that the modern contraceptive method utilization remains a great public health problem in Ethiopia. Therefore, this study aimed to assess the prevalence and associated factors of modern contraceptive utilization among married women in Ethiopia using the 2019 Ethiopia mini demographic and health survey data. 

Methods

Study settings and data source

The study used the 2019 Ethiopia mini demographic and health survey (EMDHS) dataset. It is the second mini demographic and health survey which was conducted in March, 2019, to June, 2019 in Ethiopia. Ethiopia is divided into two administrative cities and nine regions. All married women in the reproductive age living in nine regions and two administrative cities of Ethiopia were included in the study. 

Sampling procedures

The EMDHS used a complete list of 149,093 enumeration areas (EAs) created for the upcoming Ethiopia population and housing census as a sampling frame. The frame comprises information about the EA location, type of residence (urban or rural), and estimated number of residential households. Accordingly, the sample was stratified and selected in two-stages(28). 

Out of the total EAs, a total of 305 EAs (93 in urban areas and 212 in rural areas) were selected with probability proportional to EA size and with independent selection in each sampling stratum for the survey. Secondly, a fixed number of 30 households per cluster were selected with an equal probability systematic selection from the newly created household listing. 

Finally, the 2019 EMDHS survey covered 8,663 households out of the selected 8,794 households providing a response rate of 99%. About 8,885 women completed the interview from 16,583 women identified for the interview, yielding a response rate of 99%. According to the EMDHS report response rates were higher in rural than in urban areas(28). 

The source population of this study was all married non-pregnant women who were in the reproductive age group and living in Ethiopia. Pregnant mothers and those who were not in union at the time of survey were excluded from the study. Hence, 4,983 married women data were extracted from the 2019 EMDHS datasets. 

Measurements of variables

Dependent variable: Modern contraceptive utilization was the outcome variable of the study. Woman was considered as a “utilizer” if she had been utilizing any modern contraceptive methods such as oral contraceptives, male and female sterilization, intrauterine contraceptive device, injectables, implants, male and female condoms, lactational amenorrhea method, standard days method, and emergency contraception(4,30) during the 2019 EMDHS survey period while woman who had been utilizing traditional, folkloric or no method was considered as a “non-utilizer”. 

Independent Variable: Both the individual and community level explanatory variables were used to assess modern contraceptives utilized among women in childbearing age in the country.

Individual level factors: maternal age, educational level of mother, wealth index, total children ever born, number of living children, births in the last three years, age of respondent at first birth, family size, number of under five children in household, and knowledge on modern contraceptive method. 

Community-level variables: community level explanatory variables include religion, region, and place of residence (Table 1). 

Table 1: Description of individual and community level variables 

Variables Description

Individual-level variables 

Maternal age It is the current age of women recoded as 15-19, 20-24, 25-29, 30-34, 35-39, 40-44, and 45-49. 

Educational level of mother This is the level of education a woman attained and recoded as no education, primary, secondary and higher. 

Wealth index In DHS the wealth index is calculated using data on a household’s ownership of selected assets. Each household asset is assigned a weight score generated through PCA. The resulting asset scores are standardized and summed by household and individuals are ranked according to the total score of the household in which they reside. Finally, it is grouped as poorest, poorer, middle, richer and richest. It is recoded as Low Income (poorest and poorer), Middle Income and High Income (richer and richest).

Total children ever born It is recoded as 1-3 children, 4-6 children, and 7 and more

Number of living children Number of living children is recoded as no child, 1-4, 5-8, 9 and above children.

Births in the last three years This variable is recoded as no birth, one birth, and 2 and above births 

Age of respondent at 1st birth Recoded as less than 18 years, 18-24, and 25 and above years 

Family size It is number of household members and recoded as 1-4, 5-8, and 9 and above

Number of <5 children in household Number of children 5 and under in household is recoded as none, 1-2, and 3 and above children.

Knowledge on modern contraceptive method Knowledge of contraceptive method is defined as percentage who know of any method, any modern method, any traditional method and specific methods; mean number of methods known. It is recoded as “yes” if the women say they know or have heard at least one of the modern contraceptive methods and else recoded as “no”. 

Religion This variable is the religious group to which the woman associates herself and recoded as Muslim, Orthodox, Protestant, Catholic and other.

Region Region of residence is typically the first administrative level within the country, or a grouping of the first administrative level. It is grouped as Tigray, Afar, Amhara, Oromia, Somali, Benishangul, SNNPR, Gambela, Harari, Addis Ababa and Dire Dawa.

Place of residence It is the designation of the cluster or enumeration area as an urban area or a rural area.

PCA: principal components analysis

Data processing and statistical analysis

The data were extracted, cleaned, re-coded, and analyzed using STATA version 14. Descriptive statistics were presented using graphs, tables and narrations. A multilevel mixed-effect logistic regression analysis, an advanced model, was used to overcome the violation of independence of observations and equal variance assumption of the traditional logistic regression model due to a hierarchical nature of DHS data. 

We first estimated an intercept-only model or the null model (with only the outcome variable). Secondly, we included all individual level factors in the model (model I). In the third stage, we constructed model II (fitted with community-level factors only) and finally model III was fitted with both individual and community-level factors. To examine clustering and the extent to which community-level factors explain the unexplained variance of the null model, the intraclass correlation coefficient (ICC) and a proportional change in variance (PCV) were checked. The model with the lowest deviance, model III, was selected as the best-fitted model for the analysis. Variance inflation factor (VIF) was done to test the existence of multicollinearity among the independent variables.

Variables having a p-value of less than 0.2 in the bivariable analysis were selected as candidate variables for the multivariable mixed-effect logistic regression analysis. In the final model, a p-value of less than 0.05 and an adjusted odds ratio (AOR) with a 95% confidence interval (CI) were used to report statistically significant factors with modern contraceptive utilization among childbearing age women.

Ethical consideration 

This study used secondary data from demographic and health survey repositories. The survey data do not contain all identifying information. The MEASURE Demographic and Health Survey Program team allowed us to access the data upon sending an abstract of our study to an online request form http://www.measuredhsprogram.com.

Results

Socio-demographic characteristics of the study participants 

A total of 4,983 non-pregnant married women were included in the analysis. The mean age of respondents was 30.7 years (SD ± 8.29 years). From the total number of respondents, almost one quarter (23.48%) were in the age range of 25-29 years. More than half (52.26%) of the women included in the analysis had no education whereas only 314(6.30%) respondents attended higher education. In this study, women who were from low income households (41.26%) were nearly equal to women who were from high income households (42.83%). 

More than one quarter (26.73%) of the respondents had 1-3 children ever born and about 2965 (59.5%) of the women have had 1-4 number of live children. One half of the respondents (49.75%) had no birth while 352 (7.06 %) respondents had two and more births in the last three years. Out of the total respondents, one half (50.35%) were in the age range of less than 18 years whereas 842(16.9%) were in the age range of 25 and above years during their first birth. Regarding family size, 2,652(53.22%) households have had 5-8 family members and 3,088(61.97%) households have had 1 or 2 child/children aged less than 5 years. Majority of the women (93.62%) included in the survey know modern contraceptives methods.

Concerning the community-level factors, the majority (42.24%) were followers of Muslim religion and nearly three fourth (72.61%) of the women were rural residents. Slightly higher proportion of women (12.88%) were from Oromia region and the lowest proportion (6.64%) were from Addis Ababa city administration (Table 2).

Table 2: Distribution of individual and community-level factors of the married women in Ethiopia, EMDHS 2019

Individual variables Number (N) Percent (%)

Maternal age

15-19

20-24

25-29

30-34

35-39

40-44

45-49 

382

803

1170

870

810

546

402 

7.67

16.11

23.48

17.46

16.26

10.96

8.07

Educational level of mother

No education 

Primary 

Secondary 

Higher 

2604

1589

476

314 

52.26

31.89

9.55

6.30

Wealth index

Low Income

Middle Income

High Income 

2056

793

2134 

41.26

15.91

42.83

Total children ever born

1-3 children 

4-6 children 

7 and above 

2189

1462

1332 

43.93

29.34

26.73

Number of living children

No child

1-4 children

5-8 children

9 and above 

514

2965

1391

113 

10.32

59.50

27.91

2.27

Births in the last three years

No birth 

One birth

2 and above births 

2479

2152

352 

49.75

43.19

7.06

Age of respondent at first birth

Less than 18 years

18 – 24 

25 and above 

2509

1632

842 

50.35

32.75

16.90

Number of household members

1-4

5-8

9 and above 

1803

2652

528 

36.18

53.22

10.60

Number of under five children in household

None 

1-2

3 and above 

1521

3088

374 

30.52

61.97

7.51

Knowledge of any method

Knows no method

Knows only folkloric method 

Knows only traditional method

Knows modern method 

311

1

6

4665 

6.24

0.02

0.12

93.62

Religion 

Muslim 

Orthodox 

Protestant 

Catholic 

Other 

2,105

1,806

983

38

51 

42.24

36.24

19.73

0.76

1.02

Region 

Tigray 

Afar 

Amhara 

Oromia 

Somali 

Benishangul 

SNNPR

Gambela 

Harari 

Addis Ababa

Dire Dawa 

391

407

584

642

353

467

629

406

378

331

395 

7.85

8.17

11.72

12.88

7.08

9.37

12.62

8.15

7.59

6.64

7.93

Type of place of residence 

Urban 

Rural 

1365

3618 

27.39

72.61

Prevalence of modern contraceptive utilization

From the total women included in the analysis, 1,927 (38.7%) respondents utilized modern contraceptive methods. In this study, non-modern contraceptive users were those who utilized traditional methods, folkloric methods or no method at the time of the survey. Accordingly, a large number of women did not utilize any contraceptive method while the least commonly utilized method was folkloric method (Fig. 1). Furthermore, injectables were the most widely utilized modern contraceptive method (22.82%) followed by implants (9.65%), and pills (2.71%). (Fig 2). 

Fig. 1: Percentage of current use by contraceptive method type among married women in Ethiopia, EMDHS 2019 

Fig. 2: Proportion of current use by contraceptive method among married women in Ethiopia, EMDHS 2019 

Factors affecting modern contraceptive utilization 

From all independent individual and community level factors, maternal age, educational level of women, wealth index, total children ever born, births in the last three years, age of mother at first birth, number of household members, number of under five children in household, religion, region and place of residence were eligible variables for multivariable multilevel analysis.

In the multivariable multilevel analysis, the following individual-level and community-level factors were associated with modern contraceptive utilization in Ethiopia. The odds of modern contraceptive utilization was 47% [AOR=0.53; 95%CI: 0.36-0.79], 67% [AOR=0.33; 95%CI: 0.21-0.51], and 87% [AOR=0.13; 95%CI: 0.08-0.22] lower among women who aged 35-39, 40-44, and 45-49 years as compared to women aged 15-19 years. A woman who attended primary, secondary, and higher school was 1.73[AOR=1.73; 95%CI: 1.45-2.06], 1.77[AOR=1.77; 95%CI: 1.34-2.33] and 2.56[AOR=2.56; 95%CI: 1.83-3.59] times more likely to utilize modern contraceptive method as compared to woman who had not attended formal education. The odds of modern contraceptive utilization among women from households with middle and high income status were 1.72[AOR=1.72; 95%CI: 1.38-2.15] and 1.88[AOR=1.88; 95%CI: 1.51-2.34] times higher as compared to those from households with low income status. Those women who had 1-4, 5-8, and 9 and above living children were 2.51[AOR=2.51; 95%CI: 1.48-4.25], 2.67[AOR=2.67; 95%CI: 1.56-4.56] and 4.13[AOR=4.13; 95%CI: 1.94-8.80] times more likely utilize modern contraceptive method as compared to those women who had no living child. Woman who had one birth, and two and above births in the last three years was 29% [AOR=0.71; 95%CI: 0.59-0.86] and 47% [AOR=0.53; 95%CI: 0.37-0.75] lowers odds of modern contraceptive utilization as compared to those woman who had no birth in the last three years. The odds of modern contraceptive utilization was 1.8[AOR=1.80; 95%CI: 1.46-2.22] times higher among women from households where the number of under five children were 1-2 as compared to women from households where there were no under five children. 

Among community level factors, the odds of modern contraceptive method utilization was 1.86[AOR=1.86; 95%CI: 1.46-2.36] and 1.37[AOR=1.37; 95%CI: 1.02-1.83] times higher among women who have been following Orthodox religion and Protestant religion as compared to women who have been following Muslim religion. The odds of modern contraceptive method utilization among women living in Amhara, Oromia, Benishangul and Southern nation nationalities and peoples regions of Ethiopia were 2.33[AOR=2.33; 95%CI: 1.43-3.80], 2.06[AOR=2.06; 95%CI: 1.24-3.43], 1.86[AOR=1.86; 95%CI: 1.09-3.18], and 2.35[AOR=2.35; 95%CI: 1.40-3.94] times higher as compared to women living in Tigray region, respectively. The odds of modern contraceptive method utilization among women living in Afar and Somali regions were lower by 59% [AOR=0.41; 95%CI: 0.22-0.78] and 88% [AOR=0.12; 95%CI: 0.05-0.27] as compared to women living in Tigray region of Ethiopia (Table 3). 

Table 3: Multivariable multilevel analysis for factors associated with modern contraceptive utilization among married women in Ethiopia, EMDHS 2019 

Variables Null Model Model I AOR(95% CI) Model II 

AOR(95%CI) Model III AOR(95%CI)

Maternal age

15-19

20-24

25-29

30-34

35-39

40-44

45-49 

1.00

1.27[0.92-1.74]

1.01[0.73-1.41]

0.86[0.60-1.24]

0.61[0.41-0.91]

0.39[0.25-0.60]

0.15[0.09-0.25] 

1.00

1.23[0.89-1.69]

0.92[0.67-1.28]

0.77[0.54-1.11]

0.53[0.36-0.79]**

0.33[0.21-0.51]***

0.13[0.08-0.22]***

Educational level of mother

No education 

Primary 

Secondary 

Higher 

1.00

1.73[1.45-2.06] 

1.77[1.34-2.33]

2.56[1.83-3.59] 

1.00

1.62[1.35-1.93] ***

1.50[1.14-1.98]**

2.06[1.47-2.88] ***

Wealth index

Low Income

Middle Income

High Income 

1.00

1.72[1.38-2.15]

1.88[1.51-2.34] 

1.00

1.50[1.21-1.86]***

1.63[1.30-2.05]***

Total children ever born

1-3 children 

4-6 children 

7 and above 

1.00

1.13[0.90-1.42]

0.99[0.70-1.39] 

1.00

1.14[0.90-1.43]

1.01[0.72-1.42]

Number of living children

No child

1-4 children

5-8 children

9 & above 

1.00

2.57[1.52- 4.35]

2.69[1.58-4.58]

4.18[1.97-8.90] 

1.00

2.51[1.48-4.25]**

2.67[1.56-4.56]***

4.13[1.94-8.80]***

Births in the last three years

No birth 

One birth

2 and above births 

1.00

0.72[0.60-0.87]

0.47[0.33-0.67] 

1.00

0.71[0.59-0.86] ***

0.53[0.37-0.75] ***

Age of respondent at first birth

Less than 18 years

18 – 24 

25 and above 

1.00

1.24[1.04-1.47]

0.96[0.70-1.30] 

1.00

1.28[1.08-1.52]

0.93[0.68-1.26]

Number of household members

1-4

5-8

9 and above 

1.00

0.95[0.79-1.14]

0.86[0.63-1.18] 

1.00

0.94[0.79-1.13]

0.90[0.66-1.24]

Number of under five children in household

None 

1-2

3 and above 

1.00

1.79[1.45-2.22]

0.90[0.60-1.36] 

1.00

1.80[1.46-2.22] ***

1.02[0.68-1.54]

Religion 

Muslim 

Orthodox 

Protestant 

Catholic 

Other 

1.00

1.75[1.40-2.20] 

1.41[1.07-1.85]

1.80[0.81-4.01]

0.76[0.36-1.59] 

1.00

1.86[1.46-2.36] ***

1.37[1.02-1.83]

2.12[0.91-4.92]

0.92[0.42-1.98]

Region 

Tigray 

Afar 

Amhara 

Oromia 

Somali 

Benishangul 

SNNPR

Gambela 

Harari 

Addis Ababa

Dire Dawa 

1.00

0.32[0.17-0.59] 

1.82[1.13-2.93]

1.73[1.05-2.84]

0.07[0.03-0.16] 

1.57[0.93-2.66]

2.02[1.21-3.35]

1.10[0.63-1.90]

1.11[0.64-1.95]

1.48[0.83-2.66]

0.87[0.49-1.53] 

1.00

0.41[0.22-0.78]**

2.33[1.43-3.80]**

2.06[1.24-3.43]**

0.12[0.05-0.27] ***

1.86[1.09-3.18]*

2.35[1.40-3.94] **

1.07[0.61-1.88]

1.11[0.62-1.96]

1.68[0.93-3.06]

1.06[0.59-1.88]

Type of place of residence 

Urban 

Rural 

1.00

0.77[0.40-0.72] 

1.00

0.77[0.55-1.06]

*P<0.05, **P<0.001, ***P<0.0001

Regarding model fitness, as depicted in Table 4, intra-cluster correlation coefficient (ICC) of 28.97% (in the null model) confirmed the choice of multilevel model for the analysis. It indicates that 28.97% of the variation in modern contraceptive utilization among married women was due to the variation between clusters. Also the highest PCV (93.70%) in the last model (Model III) show most of the variations of modern contraceptive utilization were attributable to both individual and community level factors. Furthermore, the lowest deviance in the final model confirmed that the model (Model III) was the best fitted-model (Table 4). 

Table 4: Random effect analysis and model fitness for assessing factors associated with modern contraceptive utilization among married women in Ethiopia, EMDHS 2019

Parameters Null Model Model I Model II Model III

Community level variance (SE) 0.16 0.1458 0.0819 0.0864

ICC 0.2897 0.2453 0.1441 0.1411

PCV Reference 54.89% 92.95% 93.70

Model fitness

Log-likelihood -3039.41 -2775.74 -2927.37 -2683.76

Deviance 6078.82 5551.48 5854.74 5367.52

AIC 6082.82 5603.48 5888.75 5449.52

BIC 6095.85 5772.85 5999.48 5716.58

Discussion

 The prevalence of modern contraceptive utilization among married women in Ethiopia was 38.7% (95% CI: 37.3% to 40.0%). This finding is higher than the result of the secondary data analysis of the 2016 Ethiopia demographic and health survey where the prevalence of modern contraceptive utilization was 20.42%(11). It was also higher than the studies conducted in different parts of Ethiopia like Bale zone (20.8%), Metekel zone(18.6%), and Afar region(8.5%); and other African countries such as Ghana(21%), Burkina Faso(24%), Senegal(26.3%) and Nigeria(10.3%) (13–15,22,26,31,32). The current prevalence was higher than the national figure(35%) according to the 2016 EDHS report and the findings of the secondary data analysis of the 2016 EDHS (34.9%)(23,30). However, it was similar to a study done in Northwest Ethiopia, which was 37%(33). On the other hand, the utilization rate was lower when compared to the study conducted in Western Ethiopia (71.9%)(24), Southern Ethiopia (69.5%, 53.3%)(19,34), Northern Ethiopia (51.3%, 66.2%)(18,25), Yaoundé-Cameroon(58.9%)(35), and India(48.6%)(36). These differences could be due to participants’ cultural, religious and awareness differences on modern contraceptive methods. Additionally, variation of the study period might contributed for the observed discrepancy of modern contraceptive methods utilization.

As evidenced by the literature, the prevalence of unplanned pregnancy is lower among younger women than older women(37). This fact is supported by our study finding where the odds of modern contraceptive utilization was lower among aged women as compared to young women. This finding is also in agreement with the study findings done in Amhara regional state of Ethiopia(25), Gondar(38), East Africa(17) and findings from secondary data analysis of the 2016 EDHS(23). The possible reason might be related with the woman’s educational status. In our study, aged women were less educated as compared to the young women. Previous studies have reported that as maternal educational level increase the likelihood of desire for bearing more children decreases(39). Also there are evidences that show aged women may be in less need of modern contraceptive methods as they prefer traditional methods than the young women(40). 

Our finding also revealed that the educational level of the mother was an independent predictor of modern contraceptive method utilization among married women in Ethiopia. Women who attended primary, secondary and higher education were more likely to utilize modern contraceptive methods as compared to women with no education. This finding is in line with studies conducted in Western Ethiopia(24), Northwest Ethiopia(26,32), Ghana(12,41) and Nigeria(14). This might be due to an increase in the level of education, enhances awareness on sexual and reproductive health, maximizes probability of getting contraceptive related information and improves decision making ability of women. 

In this study, household wealth had effects on modern contraceptive method utilization where the odds of modern contraceptive utilization among married women from households with middle and high income status were higher than women from households with low income status. This finding was concordant with the findings from secondary data analysis of Ethiopia demographic and health survey(25,30), three demographic and health surveys in Bangladesh(42), a community based cross-sectional study done in Western Ethiopia(24), Eastern Ethiopia(32) and Nigeria (14). In Ethiopia, all public health facilities provide modern contraceptives for any woman in the child bearing age (15-49 years) free of charge, hence household income status has no effect in obtaining the service. However, a woman household’s wealth has great influence on exposure to education, access to basic health services and health information.

In this study, the odds of modern contraceptive utilization among women who had one or more children were higher as compared to women who had no child. This finding is supported by the study finding conducted in Southern nation nationalities and peoples region(19), North Shoa zone(43) and Senegal(15). This could be due to women who had one or more children might have greater need to space or limit births than those women who had no child. This evidence is also supported by the current study which reported that a woman from a household with one or two under five children was more likely to utilize modern contraceptive method as compared to a woman who had no child aged less than five.

This study revealed that births in the last three years were significantly associated with modern contraceptive utilization. Women who had one birth or who had two and more births were 33% and 47% times less likely to utilize modern contraceptive methods as compared to women who had no birth in the last three years respectively. The finding is consistent with the result of the study conducted in Ethiopia where modern contraceptive utilization was lower among women who had one birth or two and more births as compared to women who had no birth in the last three years (23) (43). The possible reason for this could be the variation in their desire to have more children. The problem of reverse causality might be also another reason as most of the women who reported as present utilizers may started utilizing the methods two or three years ago. 

Multivariate multilevel analysis also showed that religion of the woman was independently associated with modern contraceptive utilization. Women who had been following Orthodox religion and Protestant religion had higher odds of utilizing modern contraceptives as compared to those women who had been following Muslim religion. This finding was consistent with secondary data analysis of three demographic and health survey in Bangladesh where Muslim women have lower odds of contraceptive utilization compared with non-Muslim women(42,44). This finding was also in agreement with secondary data analysis of Ethiopia demographic and health survey in 2016 (23). 

Women’s region of residence was also associated with modern contraceptive method utilization among married women in Ethiopia. Women who live in Afar and Somali regions of Ethiopia were less likely to utilize modern contraceptive methods as compared to women who live in Tigray region. In contrast, women who reside in Amhara, Oromia, Benishangul and SNNPR regions of Ethiopia were more likely to utilize modern contraceptive methods as compared to women who live in Tigray region. The finding was in line with secondary data analysis of the 2016 Ethiopia demographic and health survey where a woman who resides in the Afar and Somali regions was nearly 55% and 93% times less likely to utilize modern contraceptive as compared to a woman who resides in Addis Ababa(11). 

The strength of the current study includes use of multilevel mixed effect analysis to overcome the hierarchical nature of EMDHS data, and deploying nationally weighted representative and most recent EMDHS data which shows country level proportion of contraceptive utilization and its associated factors among women in child bearing age. The study also has some limitations. Showing temporal relationship between modern contraceptives utilization and its predictors was impossible due to the type of the study design, cross-sectional, used for the survey. Moreover, the EMDHS data did not encompass information about some predictors of modern contraceptive utilization, as it was a mini report. Conversely, the investigators have trust that the stated limitations cannot impose significant impact on the validity of study findings. 

Conclusion 

In the current study only four out of ten married non-pregnant women of reproductive age utilized modern contraceptive methods. Furthermore, the study has identified both individual and community-level factors that can affect the utilization of modern contraceptive methods by married women in the country. Therefore, concerned bodies need to improve access to reproductive health services, empower women through community-based approaches, and minimize region wise discrepancy to optimize the utilization.

Acknowledgments

We would like to forward our kindest regards to MEASURE Demographic and Health Survey Program for providing us 2019 EMDHS data. 

Authors’ contribution

SSS designed the study, requested data, processed and analyzed data, and drafted the manuscript. TFG and BTM were equally involved in the processing, analysis, and drafting the manuscript. All authors have read and approved the final manuscript.

Abbreviations

AOR: Adjusted Odds Ration, CI: Confidence Interval, CSA: Central Statistical Agency, EMDHS: Ethiopia Mini Demographic and Health Survey, HSTP: Health Sector Transformation Plan, IUD: Interuterine Contraceptive Device, SDM: Standard Days Method 

Funding 

Not applicable

Availability of data and materials

The dataset used for the current study is available in the demographic and health survey repository in https//www.dhsprogram.com/data. 

Consent to publication

Not applicable

Competing interests

The authors declare that they haves no competing interests.

References

1. Edition T, Ababa A. Federal Democratic Republic of Ethiopia Ministry of Health Third Edition National Guideline for Family Planning Services In Ethiopia Third Edition. 2020;(July):1–77. 

2. Inter-Agency and Expert Group on Sustainable Development Goal Indicators. Report of the Inter-Agency and Expert Group on Sustainable Development Goal Indicators (E/CN.3/2016/2/Rev.1), Annex IV. Rep Inter-Agency Expert Gr Sustain Dev Goal Indic [Internet]. 2016;Annex IV. Available from: https://sustainabledevelopment.un.org/content/documents/11803Official-List-of-Proposed-SDG-Indicators.pdf

3. Hubacher D, Trussell J. A definition of modern contraceptive methods. Contraception [Internet]. 2015;92(5):420–1. Available from: http://dx.doi.org/10.1016/j.contraception.2015.08.008

4. Croft, Trevor N., Aileen M. J. Marshall, Courtney K. Allen et al. Guide to DHS Statistics. Rockville, Maryland, USA ICF. 2018;22–51. 

5. Region M. Health Gender. 2020;4–5. 

6. WHO. Adolescent pregnancy fact sheet. 2014. 

7. Schaapveld, A & Ineke V. Contraceptives Knowledge among High School Female Adolescents. Reprod Health. 2018;2(1). 

8. Kassa GM, Arowojolu AO, Odukogbe AA, Yalew AW. Prevalence and determinants of adolescent pregnancy in Africa: a systematic review and Meta-analysis. Reprod Health. 2018;15(1):1–17. 

9. Global FP. 2018 EDITION What ’ s New in This Edition ? 2018. 

10. World Fertility and Family Planning 2020: Highlights. World Fertility and Family Planning 2020: Highlights. 2020. 

11. Gebre MN, Edossa ZK. Modern contraceptive utilization and associated factors among reproductive-age women in Ethiopia: Evidence from 2016 Ethiopia demographic and health survey. BMC Womens Health. 2020;20(1):1–14. 

12. Apanga PA, Adam MA. Factors influencing the uptake of family planning services in the Talensi district, Ghana. Pan Afr Med J. 2015;20:1–9. 

13. O’Regan A, Thompson G. Indicators of young women’s modern contraceptive use in Burkina Faso and Mali from Demographic and Health Survey data. Contracept Reprod Med. 2017;2(1):1–8. 

14. Ofonime JE. Determinants of modern contraceptive uptake among Nigerian women: Evidence from the national demographic and health survey. Afr J Reprod Health [Internet]. 2017;21(3):89–95. Available from: https://www.scopus.com/inward/record.uri?eid=2-s2.0-85038842247&partnerID=40&md5=c36f097c56e5413d445dcaaf7d7a30a1

15. Zegeye B, Ahinkorah BO, Idriss-Wheeler D, Olorunsaiye CZ, Adjei NK, Yaya S. Modern contraceptive utilization and its associated factors among married women in Senegal: a multilevel analysis. BMC Public Health. 2021;21(1):1–13. 

16. Sheet F. Contraceptive use in East Africa: What do the numbers tell us? 2020;(October 2018):1–4. 

17. Tessema ZT, Teshale AB, Tesema GA, Yeshaw Y, Worku MG. Pooled prevalence and determinants of modern contraceptive utilization in East Africa: A Multi-country Analysis of recent Demographic and Health Surveys. PLoS One [Internet]. 2021;16(3 March):1–16. Available from: http://dx.doi.org/10.1371/journal.pone.0247992

18. TB Kassa GDZB. Assessment of modern contraceptive practice and associated factors among currently married women age 15-49 years in Farta District, South Gondar zone. North west Ethiop. 2015;2(6):507–12. 

19. Endriyas M, Eshete A, Mekonnen E, Misganaw T, Shiferaw M, Ayele S. Contraceptive utilization and associated factors among women of reproductive age group in Southern Nations Nationalities and Peoples’ Region, Ethiopia: cross-sectional survey, mixed-methods. Contracept Reprod Med [Internet]. 2017;2(1):1–9. Available from: http://dx.doi.org/10.1186/s40834-016-0036-z

20. Alemayehu GA, Fekadu A, Yitayal M, Kebede Y, Abebe SM, Ayele TA, et al. Prevalence and determinants of contraceptive utilization among married women at Dabat Health and Demographic Surveillance System site, northwest Ethiopia. BMC Womens Health. 2018;18(1):1–7. 

21. Debebe S, Limenih MA, Biadgo B. Modern contraceptive methods utilization and associated factors among reproductive aged women in rural Dembia District, northwest Ethiopia: Community based cross-sectional study. 2020;15(6):367–74. 

22. Beson P, Appiah R, Adomah-Afari A. Modern contraceptive use among reproductive-aged women in Ghana: Prevalence, predictors, and policy implications. BMC Womens Health. 2018;18(1):1–8. 

23. Abate MG, Tareke AA. Individual and community level associates of contraceptive use in Ethiopia: A multilevel mixed effects analysis. Arch Public Heal. 2019;77(1):1–12. 

24. Tekelab T, Melka AS, Wirtu D. Predictors of modern contraceptive methods use among married women of reproductive age groups in Western Ethiopia: A community based cross-sectional study. BMC Womens Health [Internet]. 2015;15(1):1–8. Available from: http://dx.doi.org/10.1186/s12905-015-0208-z

25. Melash Belachew Asresie , Gedefaw Abeje Fekadu, Gizachew Work Dagnew YMG. Modern Contraceptive Use and Influencing Factors in Amhara Regional State: Further Analysis of Ethiopian Demographic Health Survey Data 2016. Hindawi Adv Public Heal. 2020:8 pages. 

26. Adane AA, Bekele YA, Melese E, Worku GT, Netsere HB. Modern Contraceptive Utilization and Associated Factors among Married Gumuz Women in Metekel Zone North West Ethiopia. Biomed Res Int. 2020;2020. 

27. Andualem Samuel, Abraham Uliso, Birke Olle, Desalech Dambe MN and MMS. Assessment of Modern Contraceptive Method Utilization and Associated Factors Among Women of Reproductive Age Group in Arba Minch Town, SNNPR, Ethiopia. EC Gynaecol. 2017;6.2:36–53. 

28. Ethiopian Public Health Institute (EPHI) and ICF. 2019. Mini Demographic and Health Survey 2019: key Indicators. Rockville, Maryland, USA: EPHI and ICF. 2019. 35 p. 

29. FMOH Performance report. Federal Ministry of Health of Ethiopia Performance report 2012 EFY (2019/2020). 2020;2012:487–567. 

30. Central Statistical Agency (CSA) [Ethiopia] and ICF. Ethiopia Demographic and Health Survey 2016. Addis Ababa, Ethiop Rockville, Maryland, USA CSA ICF. 

31. Belda SS, Haile MT, Melku AT, Tololu AK. Modern contraceptive utilization and associated factors among married pastoralist women in Bale eco-region, Bale Zone, South East Ethiopia. BMC Health Serv Res. 2017;17(1):1–12. 

32. Alemayehu M, Lemma H, Abrha K, Adama Y, Fisseha G, Yebyo H, et al. Family planning use and associated factors among pastoralist community of afar region, eastern Ethiopia. BMC Womens Health [Internet]. 2016;16(1):1–9. Available from: http://dx.doi.org/10.1186/s12905-016-0321-7

33. Geremew AB, Gelagay AA. Modern contraceptive use and associated factors among married women in Finote Selam town Northwest Ethiopia: a community based cross-sectional study. Women’s Midlife Heal. 2018;4(1):1–8. 

34. A E. Contraceptive Method Mix Utilization and its Associated Factors among Married Women in Gedeo Zone, Southern Nations, Nationality and People Region-Ethiopia: A Community based Cross Sectional Study. Epidemiol Open Access. 2015;05(04). 

35. Njotang PN, Yakum MN, Ajong AB, Essi MJ, Akoh EW, Mesumbe NE, et al. Determinants of modern contraceptive practice in Yaoundé-Cameroon: A community based cross sectional study. BMC Res Notes. 2017;10(1):4–9. 

36. Dey AK. Socio-demographic determinants and modern family planning usage pattern-an analysis of National Family Health Survey-IV data. Int J Community Med Public Heal. 2019;6(2):738. 

37. Tsegaye AT, Mengistu M, Shimeka A. Prevalence of unintended pregnancy and associated factors among married women in west Belessa Woreda, Northwest Ethiopia, 2016. Reprod Health. 2018;15(1):1–8. 

38. Oumer M. Modern Contraceptive Method Utilization and Associated Factors Among Women of Reproductive Age in Gondar City , Northwest Ethiopia. 2020;53–67. 

39. Akinyemi JO, Odimegwu CO. Social contexts of fertility desire among non-childbearing young men and women aged 15–24 years in Nigeria. Reprod Health [Internet]. 2021;18(1):1–18. Available from: https://doi.org/10.1186/s12978-021-01237-1

40. Godfrey EM, Chin NP, Fielding SL, Fiscella K, Dozier A. Contraceptive methods and use by women aged 35 and over: A qualitative study of perspectives. BMC Womens Health [Internet]. 2011;11(1):5. Available from: http://www.biomedcentral.com/1472-6874/11/5

41. Aviisah PA, Dery S, Atsu BK, Yawson A, Alotaibi RM, Rezk HR, et al. Modern contraceptive use among women of reproductive age in Ghana: Analysis of the 2003-2014 Ghana Demographic and Health Surveys. BMC Womens Health. 2018;18(1):1–10. 

42. Haq I, Sakib S, Talukder A. Sociodemographic Factors on Contraceptive Use among Ever-Married Women of Reproductive Age: Evidence from Three Demographic and Health Surveys in Bangladesh. Med Sci. 2017;5(4):31. 

43. A. M, D. W, A. F, B. M. Determinants of modern contraceptive utilization among married women of reproductive age group in North Shoa Zone, Amhara Region, Ethiopia. Reprod Health [Internet]. 2014;11(1):1–7. Available from: http://ovidsp.ovid.com/ovidweb.cgi?T=JS&PAGE=reference&D=emed12&NEWS=N&AN=2014111648

44. Kibria GMA, Hossen S, Barsha RAA, Sharmeen A, Paul SK, Uddin SMI. Factors affecting contraceptive use among married women of reproductive age in Bangladesh. J Mol Stud Med Res. 2016;2(1):70–9. 

Comment: PLOS ONE does not copy edit accepted manuscripts (https://journals.plos.org/plosone/s/criteria-for-publication#loc-5). To that effect, please ensure that your submission is free of typos and grammatical errors.

Response:

Together with the language expert we have made major modifications on typos and grammatical errors of the submitted manuscript.

Four out of ten married women utilized modern contraceptive method in Ethiopia: A Multilevel analysis of the 2019 Ethiopia mini demographic and health survey 

Sewunet Sako Shagaro1*, Teshale Fikadu Gebabo2 and Be’emnet Tekabe Mulgeta1

1Department of Health Informatics, School of Public Health, College of Medicine and Health Sciences, Arba Minch University, Arba Minch, Ethiopia

2Department of Public Health, School of Public Health, College of Medicine and Health Sciences, Arba Minch University, Arba Minch, Ethiopia 

* zesew1lalem@gmail.com (SS) 

Abstract 

Background: Modern contraceptive method is a product or medical procedure that interferes with reproduction from acts of sexual intercourse. Globally in 2019, 44% of women of reproductive age were using a modern method of contraception but it was 29% in sub-Saharan Africa. Therefore, the main aim of this analysis was to assess the prevalence of modern contraceptive utilization and associated factors among married women in Ethiopia.

Method: The current study used the 2019 Ethiopia mini demographic and health survey dataset. Both descriptive and multilevel mixed-effect logistic regression analysis were done using STATA version 14. A p-value of less than 0.05 and an adjusted odds ratio with a 95% confidence interval were used to report statistically significant factors with modern contraceptive utilization.

Result: The overall modern contraceptive utilization among married women in Ethiopia was 38.7% (95% CI: 37.3% to 40.0%). Among the modern contraceptive methods, injectables were the most widely utilized modern contraceptive method (22.82%) followed by implants (9.65%) and pills (2.71%). Maternal age, educational level, wealth index, number of living children, number of births in the last three years, number of under 5 children in the household, religion, and geographic region were independent predictors of modern contraceptive utilization. 

Conclusion: In the current study only four out of ten married non-pregnant women of reproductive age utilized modern contraceptive methods. Furthermore, the study has identified both individual and community-level factors that can affect the utilization of modern contraceptive methods by married women in the country. Therefore, concerned bodies need to improve access to reproductive health services, empower women through community-based approaches, and minimize region wise discrepancy to optimize the utilization.

Keywords: modern contraceptive method, reproductive age, Ethiopia 

Introduction 

Family planning is defined as the choice of individuals or couples to anticipate and attain their desired number of children, and the spacing and timing of their births. This can be realized through use of contraceptive methods and the treatment of involuntary infertility(1). Ensuring universal access to sexual and reproductive health-care services, including for family planning, information and education, and the integration of reproductive health into national strategies and programmes; and reducing the maternal mortality ratio to less than 70 per 100,000 live births by 2030 are some of goals directly targeted to family planning mentioned under sustainable development goal (SDG) 3 of United Nations (UN)(2). 

Modern contraceptive method is a product or medical procedure that interferes with reproduction from acts of sexual intercourse(3). Modern methods of contraceptive include oral contraceptives, sterilization (male and female), interuterine contraceptive device (IUD), injectables, implants, condoms (male and female), diaphragm, lactational amenorrhea method, standard days method (SDM), emergency contraception, cervical caps, contraceptive sponges and other country specific modern methods(4). 

Literatures have shown that contraceptive utilization can save the loss of more than 3 million girls per year due to unsafe abortion. It also improves school dropout, poverty, high rate of teenage marriage and complications related to unwanted pregnancy. It is believed that access to family planning methods gives women more freedom, independence and gender equity. Moreover, contraceptives utilization plays a significant role in improving maternal and child health, women empowerment, educational advances and economic development of the country. It allows the couples to decide on the desired number of children and to space births according to their plan and income level(5–8). Therefore, realizing the rights and well-being of women can be assured by improving access to high quality, affordable sexual and reproductive health services and information mainly about the full range of family planning methods(9). 

As indicated in the World Fertility and Family Planning 2020, globally 44% of women of reproductive age were utilizing modern contraceptive methods in 2019. The use of contraception among women of child bearing age in Sub-Saharan Africa increased from 13% in 1990 to 29% in 2019(10). The report also stated that the decrease of fertility in Sub-Saharan Africa has been quite slow as compared to other regions of the world. Consequently, Sub-Saharan Africa requires 34 years (from 1995 to 2029) to decline its fertility from 6.0 to 4.0 live births per woman while it took 24 years in Eastern and South-Eastern Asia for the same reduction. Also studies conducted in various countries of Africa revealed the prevalence of modern contraceptive methods utilization among married women continues to be low(11–15). 

However, there was progress in utilization of contraceptives among women of reproductive age in the East Africa region on average from 21% (1990-94) to 39% (2010-14). The highest proportion of contraceptive users was reported from Kenya (53%) while the lowest users were from Uganda by 23% from the region(16,17). In east and other parts of Africa, injectables were the most widely utilized modern contraceptive method followed by implants(16,18–21). 

Various studies have identified maternal age, educational status, wealth index, births in the last three years, marital status, religion, region, and place of residence as individual-level and community-level factors associated with modern contraceptive utilization in Ethiopia and Africa as well(11,15,22,23). Other studies also considered husband’s education level, media exposure, number of desired children and fertility related decisions as factors that may influence contraceptive utilization(13,21,24,25). As stated in literature, higher educational status of women was positively associated with modern contraceptive utilization where the odds of modern contraceptive utilization among women who attained primary educational levels was higher than women without formal education(12–15,17,26,27). Similarly, modern contraceptive utilization was highly influenced by the wealth index of the women(11,14,15,25). 

Despite the reduction of the total fertility rate over the years, Ethiopia has still one of the highest fertility rates in Africa. The prevalence of any modern contraceptive use by currently married Ethiopian women has tripled over the last fifteen years, jumping from 14% in 2005 to 41% in 2019(28). The most commonly used contraceptive methods for currently married women in the country are injectables (27%), followed by implants (9%). The lowest contraceptive utilization was reported from the Somali region (3%) and the highest from both Amhara and Oromia regions (50%) of Ethiopia(28). However, as stated in the 2020 report of the Federal Ministry of Health of Ethiopia, the health sector transformation plan (HSTP) I target (55%) of the national contraceptive prevalence rate was not achieved. And all the regions in the country were unable to attain their target set for the year(29). These evidence clearly convey that the modern contraceptive method utilization remains a great public health problem in Ethiopia. Therefore, this study aimed to assess the prevalence and associated factors of modern contraceptive utilization among married women in Ethiopia using the 2019 Ethiopia mini demographic and health survey data. 

Methods

Study settings and data source

The study used the 2019 Ethiopia mini demographic and health survey (EMDHS) dataset. It is the second mini demographic and health survey which was conducted in March, 2019, to June, 2019 in Ethiopia. Ethiopia is divided into two administrative cities and nine regions. All married women in the reproductive age living in nine regions and two administrative cities of Ethiopia were included in the study. 

Sampling procedures

The EMDHS used a complete list of 149,093 enumeration areas (EAs) created for the upcoming Ethiopia population and housing census as a sampling frame. The frame comprises information about the EA location, type of residence (urban or rural), and estimated number of residential households. Accordingly, the sample was stratified and selected in two-stages(28). 

Out of the total EAs, a total of 305 EAs (93 in urban areas and 212 in rural areas) were selected with probability proportional to EA size and with independent selection in each sampling stratum for the survey. Secondly, a fixed number of 30 households per cluster were selected with an equal probability systematic selection from the newly created household listing. 

Finally, the 2019 EMDHS survey covered 8,663 households out of the selected 8,794 households providing a response rate of 99%. About 8,885 women completed the interview from 16,583 women identified for the interview, yielding a response rate of 99%. According to the EMDHS report response rates were higher in rural than in urban areas(28). 

The source population of this study was all married non-pregnant women who were in the reproductive age group and living in Ethiopia. Pregnant mothers and those who were not in union at the time of survey were excluded from the study. Hence, 4,983 married women data were extracted from the 2019 EMDHS datasets. 

Measurements of variables

Dependent variable: Modern contraceptive utilization was the outcome variable of the study. Woman was considered as a “utilizer” if she had been utilizing any modern contraceptive methods such as oral contraceptives, male and female sterilization, intrauterine contraceptive device, injectables, implants, male and female condoms, lactational amenorrhea method, standard days method, and emergency contraception(4,30) during the 2019 EMDHS survey period while woman who had been utilizing traditional, folkloric or no method was considered as a “non-utilizer”. 

Independent Variable: Both the individual and community level explanatory variables were used to assess modern contraceptives utilized among women in childbearing age in the country.

Individual level factors: maternal age, educational level of mother, wealth index, total children ever born, number of living children, births in the last three years, age of respondent at first birth, family size, number of under five children in household, and knowledge on modern contraceptive method. 

Community-level variables: community level explanatory variables include religion, region, and place of residence (Table 1). 

Table 1: Description of individual and community level variables 

Variables Description

Individual-level variables 

Maternal age It is the current age of women recoded as 15-19, 20-24, 25-29, 30-34, 35-39, 40-44, and 45-49. 

Educational level of mother This is the level of education a woman attained and recoded as no education, primary, secondary and higher. 

Wealth index In DHS the wealth index is calculated using data on a household’s ownership of selected assets. Each household asset is assigned a weight score generated through PCA. The resulting asset scores are standardized and summed by household and individuals are ranked according to the total score of the household in which they reside. Finally, it is grouped as poorest, poorer, middle, richer and richest. It is recoded as Low Income (poorest and poorer), Middle Income and High Income (richer and richest).

Total children ever born It is recoded as 1-3 children, 4-6 children, and 7 and more

Number of living children Number of living children is recoded as no child, 1-4, 5-8, 9 and above children.

Births in the last three years This variable is recoded as no birth, one birth, and 2 and above births 

Age of respondent at 1st birth Recoded as less than 18 years, 18-24, and 25 and above years 

Family size It is number of household members and recoded as 1-4, 5-8, and 9 and above

Number of <5 children in household Number of children 5 and under in household is recoded as none, 1-2, and 3 and above children.

Knowledge on modern contraceptive method Knowledge of contraceptive method is defined as percentage who know of any method, any modern method, any traditional method and specific methods; mean number of methods known. It is recoded as “yes” if the women say they know or have heard at least one of the modern contraceptive methods and else recoded as “no”. 

Religion This variable is the religious group to which the woman associates herself and recoded as Muslim, Orthodox, Protestant, Catholic and other.

Region Region of residence is typically the first administrative level within the country, or a grouping of the first administrative level. It is grouped as Tigray, Afar, Amhara, Oromia, Somali, Benishangul, SNNPR, Gambela, Harari, Addis Ababa and Dire Dawa.

Place of residence It is the designation of the cluster or enumeration area as an urban area or a rural area.

PCA: principal components analysis

Data processing and statistical analysis

The data were extracted, cleaned, re-coded, and analyzed using STATA version 14. Descriptive statistics were presented using graphs, tables and narrations. A multilevel mixed-effect logistic regression analysis, an advanced model, was used to overcome the violation of independence of observations and equal variance assumption of the traditional logistic regression model due to a hierarchical nature of DHS data. 

We first estimated an intercept-only model or the null model (with only the outcome variable). Secondly, we included all individual level factors in the model (model I). In the third stage, we constructed model II (fitted with community-level factors only) and finally model III was fitted with both individual and community-level factors. To examine clustering and the extent to which community-level factors explain the unexplained variance of the null model, the intraclass correlation coefficient (ICC) and a proportional change in variance (PCV) were checked. The model with the lowest deviance, model III, was selected as the best-fitted model for the analysis. Variance inflation factor (VIF) was done to test the existence of multicollinearity among the independent variables.

Variables having a p-value of less than 0.2 in the bivariable analysis were selected as candidate variables for the multivariable mixed-effect logistic regression analysis. In the final model, a p-value of less than 0.05 and an adjusted odds ratio (AOR) with a 95% confidence interval (CI) were used to report statistically significant factors with modern contraceptive utilization among childbearing age women.

Ethical consideration 

This study used secondary data from demographic and health survey repositories. The survey data do not contain all identifying information. The MEASURE Demographic and Health Survey Program team allowed us to access the data upon sending an abstract of our study to an online request form http://www.measuredhsprogram.com.

Results

Socio-demographic characteristics of the study participants 

A total of 4,983 non-pregnant married women were included in the analysis. The mean age of respondents was 30.7 years (SD ± 8.29 years). From the total number of respondents, almost one quarter (23.48%) were in the age range of 25-29 years. More than half (52.26%) of the women included in the analysis had no education whereas only 314(6.30%) respondents attended higher education. In this study, women who were from low income households (41.26%) were nearly equal to women who were from high income households (42.83%). 

More than one quarter (26.73%) of the respondents had 1-3 children ever born and about 2965 (59.5%) of the women have had 1-4 number of live children. One half of the respondents (49.75%) had no birth while 352 (7.06 %) respondents had two and more births in the last three years. Out of the total respondents, one half (50.35%) were in the age range of less than 18 years whereas 842(16.9%) were in the age range of 25 and above years during their first birth. Regarding family size, 2,652(53.22%) households have had 5-8 family members and 3,088(61.97%) households have had 1 or 2 child/children aged less than 5 years. Majority of the women (93.62%) included in the survey know modern contraceptives methods.

Concerning the community-level factors, the majority (42.24%) were followers of Muslim religion and nearly three fourth (72.61%) of the women were rural residents. Slightly higher proportion of women (12.88%) were from Oromia region and the lowest proportion (6.64%) were from Addis Ababa city administration (Table 2).

Table 2: Distribution of individual and community-level factors of the married women in Ethiopia, EMDHS 2019

Individual variables Number (N) Percent (%)

Maternal age

15-19

20-24

25-29

30-34

35-39

40-44

45-49 

382

803

1170

870

810

546

402 

7.67

16.11

23.48

17.46

16.26

10.96

8.07

Educational level of mother

No education 

Primary 

Secondary 

Higher 

2604

1589

476

314 

52.26

31.89

9.55

6.30

Wealth index

Low Income

Middle Income

High Income 

2056

793

2134 

41.26

15.91

42.83

Total children ever born

1-3 children 

4-6 children 

7 and above 

2189

1462

1332 

43.93

29.34

26.73

Number of living children

No child

1-4 children

5-8 children

9 and above 

514

2965

1391

113 

10.32

59.50

27.91

2.27

Births in the last three years

No birth 

One birth

2 and above births 

2479

2152

352 

49.75

43.19

7.06

Age of respondent at first birth

Less than 18 years

18 – 24 

25 and above 

2509

1632

842 

50.35

32.75

16.90

Number of household members

1-4

5-8

9 and above 

1803

2652

528 

36.18

53.22

10.60

Number of under five children in household

None 

1-2

3 and above 

1521

3088

374 

30.52

61.97

7.51

Knowledge of any method

Knows no method

Knows only folkloric method 

Knows only traditional method

Knows modern method 

311

1

6

4665 

6.24

0.02

0.12

93.62

Religion 

Muslim 

Orthodox 

Protestant 

Catholic 

Other 

2,105

1,806

983

38

51 

42.24

36.24

19.73

0.76

1.02

Region 

Tigray 

Afar 

Amhara 

Oromia 

Somali 

Benishangul 

SNNPR

Gambela 

Harari 

Addis Ababa

Dire Dawa 

391

407

584

642

353

467

629

406

378

331

395 

7.85

8.17

11.72

12.88

7.08

9.37

12.62

8.15

7.59

6.64

7.93

Type of place of residence 

Urban 

Rural 

1365

3618 

27.39

72.61

Prevalence of modern contraceptive utilization

From the total women included in the analysis, 1,927 (38.7%) respondents utilized modern contraceptive methods. In this study, non-modern contraceptive users were those who utilized traditional methods, folkloric methods or no method at the time of the survey. Accordingly, a large number of women did not utilize any contraceptive method while the least commonly utilized method was folkloric method (Fig. 1). Furthermore, injectables were the most widely utilized modern contraceptive method (22.82%) followed by implants (9.65%), and pills (2.71%). (Fig 2). 

Fig. 1: Percentage of current use by contraceptive method type among married women in Ethiopia, EMDHS 2019 

Fig. 2: Proportion of current use by contraceptive method among married women in Ethiopia, EMDHS 2019 

Factors affecting modern contraceptive utilization 

From all independent individual and community level factors, maternal age, educational level of women, wealth index, total children ever born, births in the last three years, age of mother at first birth, number of household members, number of under five children in household, religion, region and place of residence were eligible variables for multivariable multilevel analysis.

In the multivariable multilevel analysis, the following individual-level and community-level factors were associated with modern contraceptive utilization in Ethiopia. The odds of modern contraceptive utilization was 47% [AOR=0.53; 95%CI: 0.36-0.79], 67% [AOR=0.33; 95%CI: 0.21-0.51], and 87% [AOR=0.13; 95%CI: 0.08-0.22] lower among women who aged 35-39, 40-44, and 45-49 years as compared to women aged 15-19 years. A woman who attended primary, secondary, and higher school was 1.73[AOR=1.73; 95%CI: 1.45-2.06], 1.77[AOR=1.77; 95%CI: 1.34-2.33] and 2.56[AOR=2.56; 95%CI: 1.83-3.59] times more likely to utilize modern contraceptive method as compared to woman who had not attended formal education. The odds of modern contraceptive utilization among women from households with middle and high income status were 1.72[AOR=1.72; 95%CI: 1.38-2.15] and 1.88[AOR=1.88; 95%CI: 1.51-2.34] times higher as compared to those from households with low income status. Those women who had 1-4, 5-8, and 9 and above living children were 2.51[AOR=2.51; 95%CI: 1.48-4.25], 2.67[AOR=2.67; 95%CI: 1.56-4.56] and 4.13[AOR=4.13; 95%CI: 1.94-8.80] times more likely utilize modern contraceptive method as compared to those women who had no living child. Woman who had one birth, and two and above births in the last three years was 29% [AOR=0.71; 95%CI: 0.59-0.86] and 47% [AOR=0.53; 95%CI: 0.37-0.75] lowers odds of modern contraceptive utilization as compared to those woman who had no birth in the last three years. The odds of modern contraceptive utilization was 1.8[AOR=1.80; 95%CI: 1.46-2.22] times higher among women from households where the number of under five children were 1-2 as compared to women from households where there were no under five children. 

Among community level factors, the odds of modern contraceptive method utilization was 1.86[AOR=1.86; 95%CI: 1.46-2.36] and 1.37[AOR=1.37; 95%CI: 1.02-1.83] times higher among women who have been following Orthodox religion and Protestant religion as compared to women who have been following Muslim religion. The odds of modern contraceptive method utilization among women living in Amhara, Oromia, Benishangul and Southern nation nationalities and peoples regions of Ethiopia were 2.33[AOR=2.33; 95%CI: 1.43-3.80], 2.06[AOR=2.06; 95%CI: 1.24-3.43], 1.86[AOR=1.86; 95%CI: 1.09-3.18], and 2.35[AOR=2.35; 95%CI: 1.40-3.94] times higher as compared to women living in Tigray region, respectively. The odds of modern contraceptive method utilization among women living in Afar and Somali regions were lower by 59% [AOR=0.41; 95%CI: 0.22-0.78] and 88% [AOR=0.12; 95%CI: 0.05-0.27] as compared to women living in Tigray region of Ethiopia (Table 3). 

Table 3: Multivariable multilevel analysis for factors associated with modern contraceptive utilization among married women in Ethiopia, EMDHS 2019 

Variables Null Model Model I AOR(95% CI) Model II 

AOR(95%CI) Model III AOR(95%CI)

Maternal age

15-19

20-24

25-29

30-34

35-39

40-44

45-49 

1.00

1.27[0.92-1.74]

1.01[0.73-1.41]

0.86[0.60-1.24]

0.61[0.41-0.91]

0.39[0.25-0.60]

0.15[0.09-0.25] 

1.00

1.23[0.89-1.69]

0.92[0.67-1.28]

0.77[0.54-1.11]

0.53[0.36-0.79]**

0.33[0.21-0.51]***

0.13[0.08-0.22]***

Educational level of mother

No education 

Primary 

Secondary 

Higher 

1.00

1.73[1.45-2.06] 

1.77[1.34-2.33]

2.56[1.83-3.59] 

1.00

1.62[1.35-1.93] ***

1.50[1.14-1.98]**

2.06[1.47-2.88] ***

Wealth index

Low Income

Middle Income

High Income 

1.00

1.72[1.38-2.15]

1.88[1.51-2.34] 

1.00

1.50[1.21-1.86]***

1.63[1.30-2.05]***

Total children ever born

1-3 children 

4-6 children 

7 and above 

1.00

1.13[0.90-1.42]

0.99[0.70-1.39] 

1.00

1.14[0.90-1.43]

1.01[0.72-1.42]

Number of living children

No child

1-4 children

5-8 children

9 & above 

1.00

2.57[1.52- 4.35]

2.69[1.58-4.58]

4.18[1.97-8.90] 

1.00

2.51[1.48-4.25]**

2.67[1.56-4.56]***

4.13[1.94-8.80]***

Births in the last three years

No birth 

One birth

2 and above births 

1.00

0.72[0.60-0.87]

0.47[0.33-0.67] 

1.00

0.71[0.59-0.86] ***

0.53[0.37-0.75] ***

Age of respondent at first birth

Less than 18 years

18 – 24 

25 and above 

1.00

1.24[1.04-1.47]

0.96[0.70-1.30] 

1.00

1.28[1.08-1.52]

0.93[0.68-1.26]

Number of household members

1-4

5-8

9 and above 

1.00

0.95[0.79-1.14]

0.86[0.63-1.18] 

1.00

0.94[0.79-1.13]

0.90[0.66-1.24]

Number of under five children in household

None 

1-2

3 and above 

1.00

1.79[1.45-2.22]

0.90[0.60-1.36] 

1.00

1.80[1.46-2.22] ***

1.02[0.68-1.54]

Religion 

Muslim 

Orthodox 

Protestant 

Catholic 

Other 

1.00

1.75[1.40-2.20] 

1.41[1.07-1.85]

1.80[0.81-4.01]

0.76[0.36-1.59] 

1.00

1.86[1.46-2.36] ***

1.37[1.02-1.83]

2.12[0.91-4.92]

0.92[0.42-1.98]

Region 

Tigray 

Afar 

Amhara 

Oromia 

Somali 

Benishangul 

SNNPR

Gambela 

Harari 

Addis Ababa

Dire Dawa 

1.00

0.32[0.17-0.59] 

1.82[1.13-2.93]

1.73[1.05-2.84]

0.07[0.03-0.16] 

1.57[0.93-2.66]

2.02[1.21-3.35]

1.10[0.63-1.90]

1.11[0.64-1.95]

1.48[0.83-2.66]

0.87[0.49-1.53] 

1.00

0.41[0.22-0.78]**

2.33[1.43-3.80]**

2.06[1.24-3.43]**

0.12[0.05-0.27] ***

1.86[1.09-3.18]*

2.35[1.40-3.94] **

1.07[0.61-1.88]

1.11[0.62-1.96]

1.68[0.93-3.06]

1.06[0.59-1.88]

Type of place of residence 

Urban 

Rural 

1.00

0.77[0.40-0.72] 

1.00

0.77[0.55-1.06]

*P<0.05, **P<0.001, ***P<0.0001

Regarding model fitness, as depicted in Table 4, intra-cluster correlation coefficient (ICC) of 28.97% (in the null model) confirmed the choice of multilevel model for the analysis. It indicates that 28.97% of the variation in modern contraceptive utilization among married women was due to the variation between clusters. Also the highest PCV (93.70%) in the last model (Model III) show most of the variations of modern contraceptive utilization were attributable to both individual and community level factors. Furthermore, the lowest deviance in the final model confirmed that the model (Model III) was the best fitted-model (Table 4). 

Table 4: Random effect analysis and model fitness for assessing factors associated with modern contraceptive utilization among married women in Ethiopia, EMDHS 2019

Parameters Null Model Model I Model II Model III

Community level variance (SE) 0.16 0.1458 0.0819 0.0864

ICC 0.2897 0.2453 0.1441 0.1411

PCV Reference 54.89% 92.95% 93.70

Model fitness

Log-likelihood -3039.41 -2775.74 -2927.37 -2683.76

Deviance 6078.82 5551.48 5854.74 5367.52

AIC 6082.82 5603.48 5888.75 5449.52

BIC 6095.85 5772.85 5999.48 5716.58

Discussion

 The prevalence of modern contraceptive utilization among married women in Ethiopia was 38.7% (95% CI: 37.3% to 40.0%). This finding is higher than the result of the secondary data analysis of the 2016 Ethiopia demographic and health survey where the prevalence of modern contraceptive utilization was 20.42%(11). It was also higher than the studies conducted in different parts of Ethiopia like Bale zone (20.8%), Metekel zone(18.6%), and Afar region(8.5%); and other African countries such as Ghana(21%), Burkina Faso(24%), Senegal(26.3%) and Nigeria(10.3%) (13–15,22,26,31,32). The current prevalence was higher than the national figure(35%) according to the 2016 EDHS report and the findings of the secondary data analysis of the 2016 EDHS (34.9%)(23,30). However, it was similar to a study done in Northwest Ethiopia, which was 37%(33). On the other hand, the utilization rate was lower when compared to the study conducted in Western Ethiopia (71.9%)(24), Southern Ethiopia (69.5%, 53.3%)(19,34), Northern Ethiopia (51.3%, 66.2%)(18,25), Yaoundé-Cameroon(58.9%)(35), and India(48.6%)(36). These differences could be due to participants’ cultural, religious and awareness differences on modern contraceptive methods. Additionally, variation of the study period might contributed for the observed discrepancy of modern contraceptive methods utilization.

As evidenced by the literature, the prevalence of unplanned pregnancy is lower among younger women than older women(37). This fact is supported by our study finding where the odds of modern contraceptive utilization was lower among aged women as compared to young women. This finding is also in agreement with the study findings done in Amhara regional state of Ethiopia(25), Gondar(38), East Africa(17) and findings from secondary data analysis of the 2016 EDHS(23). The possible reason might be related with the woman’s educational status. In our study, aged women were less educated as compared to the young women. Previous studies have reported that as maternal educational level increase the likelihood of desire for bearing more children decreases(39). Also there are evidences that show aged women may be in less need of modern contraceptive methods as they prefer traditional methods than the young women(40). 

Our finding also revealed that the educational level of the mother was an independent predictor of modern contraceptive method utilization among married women in Ethiopia. Women who attended primary, secondary and higher education were more likely to utilize modern contraceptive methods as compared to women with no education. This finding is in line with studies conducted in Western Ethiopia(24), Northwest Ethiopia(26,32), Ghana(12,41) and Nigeria(14). This might be due to an increase in the level of education, enhances awareness on sexual and reproductive health, maximizes probability of getting contraceptive related information and improves decision making ability of women. 

In this study, household wealth had effects on modern contraceptive method utilization where the odds of modern contraceptive utilization among married women from households with middle and high income status were higher than women from households with low income status. This finding was concordant with the findings from secondary data analysis of Ethiopia demographic and health survey(25,30), three demographic and health surveys in Bangladesh(42), a community based cross-sectional study done in Western Ethiopia(24), Eastern Ethiopia(32) and Nigeria (14). In Ethiopia, all public health facilities provide modern contraceptives for any woman in the child bearing age (15-49 years) free of charge, hence household income status has no effect in obtaining the service. However, a woman household’s wealth has great influence on exposure to education, access to basic health services and health information.

In this study, the odds of modern contraceptive utilization among women who had one or more children were higher as compared to women who had no child. This finding is supported by the study finding conducted in Southern nation nationalities and peoples region(19), North Shoa zone(43) and Senegal(15). This could be due to women who had one or more children might have greater need to space or limit births than those women who had no child. This evidence is also supported by the current study which reported that a woman from a household with one or two under five children was more likely to utilize modern contraceptive method as compared to a woman who had no child aged less than five.

This study revealed that births in the last three years were significantly associated with modern contraceptive utilization. Women who had one birth or who had two and more births were 33% and 47% times less likely to utilize modern contraceptive methods as compared to women who had no birth in the last three years respectively. The finding is consistent with the result of the study conducted in Ethiopia where modern contraceptive utilization was lower among women who had one birth or two and more births as compared to women who had no birth in the last three years (23) (43). The possible reason for this could be the variation in their desire to have more children. The problem of reverse causality might be also another reason as most of the women who reported as present utilizers may started utilizing the methods two or three years ago. 

Multivariate multilevel analysis also showed that religion of the woman was independently associated with modern contraceptive utilization. Women who had been following Orthodox religion and Protestant religion had higher odds of utilizing modern contraceptives as compared to those women who had been following Muslim religion. This finding was consistent with secondary data analysis of three demographic and health survey in Bangladesh where Muslim women have lower odds of contraceptive utilization compared with non-Muslim women(42,44). This finding was also in agreement with secondary data analysis of Ethiopia demographic and health survey in 2016 (23). 

Women’s region of residence was also associated with modern contraceptive method utilization among married women in Ethiopia. Women who live in Afar and Somali regions of Ethiopia were less likely to utilize modern contraceptive methods as compared to women who live in Tigray region. In contrast, women who reside in Amhara, Oromia, Benishangul and SNNPR regions of Ethiopia were more likely to utilize modern contraceptive methods as compared to women who live in Tigray region. The finding was in line with secondary data analysis of the 2016 Ethiopia demographic and health survey where a woman who resides in the Afar and Somali regions was nearly 55% and 93% times less likely to utilize modern contraceptive as compared to a woman who resides in Addis Ababa(11). 

The strength of the current study includes use of multilevel mixed effect analysis to overcome the hierarchical nature of EMDHS data, and deploying nationally weighted representative and most recent EMDHS data which shows country level proportion of contraceptive utilization and its associated factors among women in child bearing age. The study also has some limitations. Showing temporal relationship between modern contraceptives utilization and its predictors was impossible due to the type of the study design, cross-sectional, used for the survey. Moreover, the EMDHS data did not encompass information about some predictors of modern contraceptive utilization, as it was a mini report. Conversely, the investigators have trust that the stated limitations cannot impose significant impact on the validity of study findings. 

Conclusion 

In the current study only four out of ten married non-pregnant women of reproductive age utilized modern contraceptive methods. Furthermore, the study has identified both individual and community-level factors that can affect the utilization of modern contraceptive methods by married women in the country. Therefore, concerned bodies need to improve access to reproductive health services, empower women through community-based approaches, and minimize region wise discrepancy to optimize the utilization.

Acknowledgments

We would like to forward our kindest regards to MEASURE Demographic and Health Survey Program for providing us 2019 EMDHS data. 

Authors’ contribution

SSS designed the study, requested data, processed and analyzed data, and drafted the manuscript. TFG and BTM were equally involved in the processing, analysis, and drafting the manuscript. All authors have read and approved the final manuscript.

Abbreviations

AOR: Adjusted Odds Ration, CI: Confidence Interval, CSA: Central Statistical Agency, EMDHS: Ethiopia Mini Demographic and Health Survey, HSTP: Health Sector Transformation Plan, IUD: Interuterine Contraceptive Device, SDM: Standard Days Method 

Funding 

Not applicable

Availability of data and materials

The dataset used for the current study is available in the demographic and health survey repository in https//www.dhsprogram.com/data. 

Consent to publication

Not applicable

Competing interests

The authors declare that they haves no competing interests.

References

1. Edition T, Ababa A. Federal Democratic Republic of Ethiopia Ministry of Health Third Edition National Guideline for Family Planning Services In Ethiopia Third Edition. 2020;(July):1–77. 

2. Inter-Agency and Expert Group on Sustainable Development Goal Indicators. Report of the Inter-Agency and Expert Group on Sustainable Development Goal Indicators (E/CN.3/2016/2/Rev.1), Annex IV. Rep Inter-Agency Expert Gr Sustain Dev Goal Indic [Internet]. 2016;Annex IV. Available from: https://sustainabledevelopment.un.org/content/documents/11803Official-List-of-Proposed-SDG-Indicators.pdf

3. Hubacher D, Trussell J. A definition of modern contraceptive methods. Contraception [Internet]. 2015;92(5):420–1. Available from: http://dx.doi.org/10.1016/j.contraception.2015.08.008

4. Croft, Trevor N., Aileen M. J. Marshall, Courtney K. Allen et al. Guide to DHS Statistics. Rockville, Maryland, USA ICF. 2018;22–51. 

5. Region M. Health Gender. 2020;4–5. 

6. WHO. Adolescent pregnancy fact sheet. 2014. 

7. Schaapveld, A & Ineke V. Contraceptives Knowledge among High School Female Adolescents. Reprod Health. 2018;2(1). 

8. Kassa GM, Arowojolu AO, Odukogbe AA, Yalew AW. Prevalence and determinants of adolescent pregnancy in Africa: a systematic review and Meta-analysis. Reprod Health. 2018;15(1):1–17. 

9. Global FP. 2018 EDITION What ’ s New in This Edition ? 2018. 

10. World Fertility and Family Planning 2020: Highlights. World Fertility and Family Planning 2020: Highlights. 2020. 

11. Gebre MN, Edossa ZK. Modern contraceptive utilization and associated factors among reproductive-age women in Ethiopia: Evidence from 2016 Ethiopia demographic and health survey. BMC Womens Health. 2020;20(1):1–14. 

12. Apanga PA, Adam MA. Factors influencing the uptake of family planning services in the Talensi district, Ghana. Pan Afr Med J. 2015;20:1–9. 

13. O’Regan A, Thompson G. Indicators of young women’s modern contraceptive use in Burkina Faso and Mali from Demographic and Health Survey data. Contracept Reprod Med. 2017;2(1):1–8. 

14. Ofonime JE. Determinants of modern contraceptive uptake among Nigerian women: Evidence from the national demographic and health survey. Afr J Reprod Health [Internet]. 2017;21(3):89–95. Available from: https://www.scopus.com/inward/record.uri?eid=2-s2.0-85038842247&partnerID=40&md5=c36f097c56e5413d445dcaaf7d7a30a1

15. Zegeye B, Ahinkorah BO, Idriss-Wheeler D, Olorunsaiye CZ, Adjei NK, Yaya S. Modern contraceptive utilization and its associated factors among married women in Senegal: a multilevel analysis. BMC Public Health. 2021;21(1):1–13. 

16. Sheet F. Contraceptive use in East Africa: What do the numbers tell us? 2020;(October 2018):1–4. 

17. Tessema ZT, Teshale AB, Tesema GA, Yeshaw Y, Worku MG. Pooled prevalence and determinants of modern contraceptive utilization in East Africa: A Multi-country Analysis of recent Demographic and Health Surveys. PLoS One [Internet]. 2021;16(3 March):1–16. Available from: http://dx.doi.org/10.1371/journal.pone.0247992

18. TB Kassa GDZB. Assessment of modern contraceptive practice and associated factors among currently married women age 15-49 years in Farta District, South Gondar zone. North west Ethiop. 2015;2(6):507–12. 

19. Endriyas M, Eshete A, Mekonnen E, Misganaw T, Shiferaw M, Ayele S. Contraceptive utilization and associated factors among women of reproductive age group in Southern Nations Nationalities and Peoples’ Region, Ethiopia: cross-sectional survey, mixed-methods. Contracept Reprod Med [Internet]. 2017;2(1):1–9. Available from: http://dx.doi.org/10.1186/s40834-016-0036-z

20. Alemayehu GA, Fekadu A, Yitayal M, Kebede Y, Abebe SM, Ayele TA, et al. Prevalence and determinants of contraceptive utilization among married women at Dabat Health and Demographic Surveillance System site, northwest Ethiopia. BMC Womens Health. 2018;18(1):1–7. 

21. Debebe S, Limenih MA, Biadgo B. Modern contraceptive methods utilization and associated factors among reproductive aged women in rural Dembia District, northwest Ethiopia: Community based cross-sectional study. 2020;15(6):367–74. 

22. Beson P, Appiah R, Adomah-Afari A. Modern contraceptive use among reproductive-aged women in Ghana: Prevalence, predictors, and policy implications. BMC Womens Health. 2018;18(1):1–8. 

23. Abate MG, Tareke AA. Individual and community level associates of contraceptive use in Ethiopia: A multilevel mixed effects analysis. Arch Public Heal. 2019;77(1):1–12. 

24. Tekelab T, Melka AS, Wirtu D. Predictors of modern contraceptive methods use among married women of reproductive age groups in Western Ethiopia: A community based cross-sectional study. BMC Womens Health [Internet]. 2015;15(1):1–8. Available from: http://dx.doi.org/10.1186/s12905-015-0208-z

25. Melash Belachew Asresie , Gedefaw Abeje Fekadu, Gizachew Work Dagnew YMG. Modern Contraceptive Use and Influencing Factors in Amhara Regional State: Further Analysis of Ethiopian Demographic Health Survey Data 2016. Hindawi Adv Public Heal. 2020:8 pages. 

26. Adane AA, Bekele YA, Melese E, Worku GT, Netsere HB. Modern Contraceptive Utilization and Associated Factors among Married Gumuz Women in Metekel Zone North West Ethiopia. Biomed Res Int. 2020;2020. 

27. Andualem Samuel, Abraham Uliso, Birke Olle, Desalech Dambe MN and MMS. Assessment of Modern Contraceptive Method Utilization and Associated Factors Among Women of Reproductive Age Group in Arba Minch Town, SNNPR, Ethiopia. EC Gynaecol. 2017;6.2:36–53. 

28. Ethiopian Public Health Institute (EPHI) and ICF. 2019. Mini Demographic and Health Survey 2019: key Indicators. Rockville, Maryland, USA: EPHI and ICF. 2019. 35 p. 

29. FMOH Performance report. Federal Ministry of Health of Ethiopia Performance report 2012 EFY (2019/2020). 2020;2012:487–567. 

30. Central Statistical Agency (CSA) [Ethiopia] and ICF. Ethiopia Demographic and Health Survey 2016. Addis Ababa, Ethiop Rockville, Maryland, USA CSA ICF. 

31. Belda SS, Haile MT, Melku AT, Tololu AK. Modern contraceptive utilization and associated factors among married pastoralist women in Bale eco-region, Bale Zone, South East Ethiopia. BMC Health Serv Res. 2017;17(1):1–12. 

32. Alemayehu M, Lemma H, Abrha K, Adama Y, Fisseha G, Yebyo H, et al. Family planning use and associated factors among pastoralist community of afar region, eastern Ethiopia. BMC Womens Health [Internet]. 2016;16(1):1–9. Available from: http://dx.doi.org/10.1186/s12905-016-0321-7

33. Geremew AB, Gelagay AA. Modern contraceptive use and associated factors among married women in Finote Selam town Northwest Ethiopia: a community based cross-sectional study. Women’s Midlife Heal. 2018;4(1):1–8. 

34. A E. Contraceptive Method Mix Utilization and its Associated Factors among Married Women in Gedeo Zone, Southern Nations, Nationality and People Region-Ethiopia: A Community based Cross Sectional Study. Epidemiol Open Access. 2015;05(04). 

35. Njotang PN, Yakum MN, Ajong AB, Essi MJ, Akoh EW, Mesumbe NE, et al. Determinants of modern contraceptive practice in Yaoundé-Cameroon: A community based cross sectional study. BMC Res Notes. 2017;10(1):4–9. 

36. Dey AK. Socio-demographic determinants and modern family planning usage pattern-an analysis of National Family Health Survey-IV data. Int J Community Med Public Heal. 2019;6(2):738. 

37. Tsegaye AT, Mengistu M, Shimeka A. Prevalence of unintended pregnancy and associated factors among married women in west Belessa Woreda, Northwest Ethiopia, 2016. Reprod Health. 2018;15(1):1–8. 

38. Oumer M. Modern Contraceptive Method Utilization and Associated Factors Among Women of Reproductive Age in Gondar City , Northwest Ethiopia. 2020;53–67. 

39. Akinyemi JO, Odimegwu CO. Social contexts of fertility desire among non-childbearing young men and women aged 15–24 years in Nigeria. Reprod Health [Internet]. 2021;18(1):1–18. Available from: https://doi.org/10.1186/s12978-021-01237-1

40. Godfrey EM, Chin NP, Fielding SL, Fiscella K, Dozier A. Contraceptive methods and use by women aged 35 and over: A qualitative study of perspectives. BMC Womens Health [Internet]. 2011;11(1):5. Available from: http://www.biomedcentral.com/1472-6874/11/5

41. Aviisah PA, Dery S, Atsu BK, Yawson A, Alotaibi RM, Rezk HR, et al. Modern contraceptive use among women of reproductive age in Ghana: Analysis of the 2003-2014 Ghana Demographic and Health Surveys. BMC Womens Health. 2018;18(1):1–10. 

42. Haq I, Sakib S, Talukder A. Sociodemographic Factors on Contraceptive Use among Ever-Married Women of Reproductive Age: Evidence from Three Demographic and Health Surveys in Bangladesh. Med Sci. 2017;5(4):31. 

43. A. M, D. W, A. F, B. M. Determinants of modern contraceptive utilization among married women of reproductive age group in North Shoa Zone, Amhara Region, Ethiopia. Reprod Health [Internet]. 2014;11(1):1–7. Available from: http://ovidsp.ovid.com/ovidweb.cgi?T=JS&PAGE=reference&D=emed12&NEWS=N&AN=2014111648

44. Kibria GMA, Hossen S, Barsha RAA, Sharmeen A, Paul SK, Uddin SMI. Factors affecting contraceptive use among married women of reproductive age in Bangladesh. J Mol Stud Med Res. 2016;2(1):70–9. 

Comment:

Your ethics statement should only appear in the Methods section of your manuscript. If your ethics statement is written in any section besides the Methods, please move it to the Methods section and delete it from any other section. Please ensure that your ethics statement is included in your manuscript, as the ethics statement entered into the online submission form will not be published alongside your manuscript.

Response:

Ethical consideration was included under method section 

Lines 158-162

Ethical consideration 

This study used secondary data from demographic and health survey repositories. The survey data do not contain all identifying information. The MEASURE Demographic and Health Survey Program team allowed us to access the data upon sending an abstract of our study to an online request form http://www.measuredhsprogram.com.

Comment:

We note you have included a table to which you do not refer in the text of your manuscript. Please ensure that you refer to Table 1 in your text; if accepted, production will need this reference to link the reader to the Table.

Response:

Now we have mentioned and placed Table 1 immediately after where table 1 is referred in the text of our manuscript

Lines 134 and 135

Community-level variables: community level explanatory variables include religion, region, and place of residence (Table 1). 

Reviewer #1

Comment:

This study aims to assess the prevalence and associated factors of modern contraceptive utilization among married women in Ethiopia using the 2019 Ethiopia Mini Demographic and Health Survey data. Findings identify a number of significant relationships between independent and outcome variables that conform to other studies in the region, while indicators on prevalence differ from other country studies.

The study approach merits the importance of utilizing secondary datasets for analyzing in more depth DHS data and particularly the application of multivariate analysis in identifying key variables for further investigation. Noted is the higher contraception rate identified in your analysis to that reported in other studies. Could you clarify if this was due to the time at which these studies were conducted or was the variance in findings across studies due to framing of questions of studies which identified lower levels lower levels of contraceptive use?

Response:

According to the 2019 Ethiopian mini demographic and health survey (EMDHS) report, the modern contraceptive methods utilization among currently married Ethiopian women was 41%(1). Hence, the prevalence of modern contraceptive methods utilization of this study is in line with this report (it is in between the upper and lower CI of the 2019 EMDHS prevalence). 

And also we have tried to discuss the trends of modern contraceptive utilization in the country over the last 15 years. 

Lines 84-87

Despite the reduction of the total fertility rate over the years, Ethiopia has still one of the highest fertility rates in Africa. The prevalence of any modern contraceptive use by currently married Ethiopian women has tripled over the last fifteen years, jumping from 14% in 2005 to 41% in 2019(28).

Additionally, the effect of variation of the study period in relation to the previous studies were also discussed. 

 Lines 251-259

The current prevalence was higher than the national figure(35%) according to the 2016 EDHS report and the findings of the secondary data analysis of the 2016 EDHS (34.9%)(23,30). However, it was similar to a study done in Northwest Ethiopia, which was 37%(33). On the other hand, the utilization rate was lower when compared to the study conducted in Western Ethiopia (71.9%)(24), Southern Ethiopia (69.5%, 53.3%)(19,34), Northern Ethiopia (51.3%, 66.2%)(18,25), Yaoundé-Cameroon(58.9%)(35), and India(48.6%)(36). These differences could be due to participants’ cultural, religious and awareness differences on modern contraceptive methods. Additionally, variation of the study period might contributed for the observed discrepancy of modern contraceptive methods utilization.

Comment:

Conclusions are supported by other regional and international findings in the area but could be more fully examined in terms of their implications for future interventions to further build contraceptive use in Ethiopia.

Response:

We have tried to re-write the conclusion section of this study.

Line 335-337

Therefore, concerned bodies need to improve access to reproductive health services, empower women through community-based approaches, and minimize region wise discrepancy to optimize the utilization.

Comment:

Minor comments is that the manuscript would benefit from editing for grammar. Eg: “those women who were born children” Does it mean "those women who had borne children? 

Response:

We have corrected the statement according to the given comment.

Lines 214-217

Woman who had one birth, and two and above births in the last three years was 29% [AOR=0.71; 95%CI: 0.59-0.86] and 47% [AOR=0.53; 95%CI: 0.37-0.75] lowers odds of modern contraceptive utilization as compared to those woman who had no birth in the last three years.

Comment:

“First wealth quantile” Does this mean highest wealth quintile? “Poorest and Richest” are loaded terms - perhaps use “Low Income” Low -to Middle Income”, ”Middle Income”, and “High Income” as per the income classifications in other national surveys.

Response:

We have also recoded the ‘wealth index’ by merging poorest and poorer as “low income”, richer and richest as “high income”, and by keeping “medium” as it is based on your comment and other national survey classifications. Then, we have done analysis and modified the whole result and discussion sections of the submitted manuscript. 

Lines 136-137

Table 1: Description of individual and community level variables 

Variables Description

Individual-level variables 

Maternal age It is the current age of women recoded as 15-19, 20-24, 25-29, 30-34, 35-39, 40-44, and 45-49. 

Educational level of mother This is the level of education a woman attained and recoded as no education, primary, secondary and higher. 

Wealth index In DHS the wealth index is calculated using data on a household’s ownership of selected assets. Each household asset is assigned a weight score generated through PCA. The resulting asset scores are standardized and summed by household and individuals are ranked according to the total score of the household in which they reside. Finally, it is grouped as poorest, poorer, middle, richer and richest. It is recoded as Low Income (poorest and poorer), Middle Income and High Income (richer and richest).

Total children ever born It is recoded as 1-3 children, 4-6 children, and 7 and more

Number of living children Number of living children is recoded as no child, 1-4, 5-8, 9 and above children.

Births in the last three years This variable is recoded as no birth, one birth, and 2 and above births 

Age of respondent at 1st birth Recoded as less than 18 years, 18-24, and 25 and above years 

Family size It is number of household members and recoded as 1-4, 5-8, and 9 and above

Number of <5 children in household Number of children 5 and under in household is recoded as none, 1-2, and 3 and above children.

Knowledge on modern contraceptive method Knowledge of contraceptive method is defined as percentage who know of any method, any modern method, any traditional method and specific methods; mean number of methods known. It is recoded as “yes” if the women say they know or have heard at least one of the modern contraceptive methods and else recoded as “no”. 

Religion This variable is the religious group to which the woman associates herself and recoded as Muslim, Orthodox, Protestant, Catholic and other.

Region Region of residence is typically the first administrative level within the country, or a grouping of the first administrative level. It is grouped as Tigray, Afar, Amhara, Oromia, Somali, Benishangul, SNNPR, Gambela, Harari, Addis Ababa and Dire Dawa.

Place of residence It is the designation of the cluster or enumeration area as an urban area or a rural area.

PCA: principal components analysis

Lines 169 and 170

In this study, women who were from low income households (41.26%) were nearly equal to women who were from high income households (42.83%). 

Lines 183 and184

Table 2: Distribution of individual and community-level factors of the married women in Ethiopia, EMDHS 2019

Individual variables Number (N) Percent (%)

Maternal age

15-19

20-24

25-29

30-34

35-39

40-44

45-49 

382

803

1170

870

810

546

402 

7.67

16.11

23.48

17.46

16.26

10.96

8.07

Educational level of mother

No education 

Primary 

Secondary 

Higher 

2604

1589

476

314 

52.26

31.89

9.55

6.30

Wealth index

Low Income

Middle Income

High Income 

2056

793

2134 

41.26

15.91

42.83

Total children ever born

1-3 children 

4-6 children 

7 and above 

2189

1462

1332 

43.93

29.34

26.73

Number of living children

No child

1-4 children

5-8 children

9 and above 

514

2965

1391

113 

10.32

59.50

27.91

2.27

Births in the last three years

No birth 

One birth

2 and above births 

2479

2152

352 

49.75

43.19

7.06

Age of respondent at first birth

Less than 18 years

18 – 24 

25 and above 

2509

1632

842 

50.35

32.75

16.90

Number of household members

1-4

5-8

9 and above 

1803

2652

528 

36.18

53.22

10.60

Number of under five children in household

None 

1-2

3 and above 

1521

3088

374 

30.52

61.97

7.51

Knowledge of any method

Knows no method

Knows only folkloric method 

Knows only traditional method

Knows modern method 

311

1

6

4665 

6.24

0.02

0.12

93.62

Religion 

Muslim 

Orthodox 

Protestant 

Catholic 

Other 

2,105

1,806

983

38

51 

42.24

36.24

19.73

0.76

1.02

Region 

Tigray 

Afar 

Amhara 

Oromia 

Somali 

Benishangul 

SNNPR

Gambela 

Harari 

Addis Ababa

Dire Dawa 

391

407

584

642

353

467

629

406

378

331

395 

7.85

8.17

11.72

12.88

7.08

9.37

12.62

8.15

7.59

6.64

7.93

Type of place of residence 

Urban 

Rural 

1365

3618 

27.39

72.61

Lines 209-211

The odds of modern contraceptive utilization among women from households with middle and high income status were 1.72[AOR=1.72; 95%CI: 1.38-2.15] and 1.88[AOR=1.88; 95%CI: 1.51-2.34] times higher as compared to those from households with low income status.

Lines 232-234

Table 3: Multivariable multilevel analysis for factors associated with modern contraceptive utilization among married women in Ethiopia, EMDHS 2019 

Variables Null Model Model I AOR(95% CI) Model II 

AOR(95%CI) Model III AOR(95%CI)

Maternal age

15-19

20-24

25-29

30-34

35-39

40-44

45-49 

1.00

1.27[0.92-1.74]

1.01[0.73-1.41]

0.86[0.60-1.24]

0.61[0.41-0.91]

0.39[0.25-0.60]

0.15[0.09-0.25] 

1.00

1.23[0.89-1.69]

0.92[0.67-1.28]

0.77[0.54-1.11]

0.53[0.36-0.79]**

0.33[0.21-0.51]***

0.13[0.08-0.22]***

Educational level of mother

No education 

Primary 

Secondary 

Higher 

1.00

1.73[1.45-2.06] 

1.77[1.34-2.33]

2.56[1.83-3.59] 

1.00

1.62[1.35-1.93] ***

1.50[1.14-1.98]**

2.06[1.47-2.88] ***

Wealth index

Low Income

Middle Income

High Income 

1.00

1.72[1.38-2.15]

1.88[1.51-2.34] 

1.00

1.50[1.21-1.86]***

1.63[1.30-2.05]***

Total children ever born

1-3 children 

4-6 children 

7 and above 

1.00

1.13[0.90-1.42]

0.99[0.70-1.39] 

1.00

1.14[0.90-1.43]

1.01[0.72-1.42]

Number of living children

No child

1-4 children

5-8 children

9 & above 

1.00

2.57[1.52- 4.35]

2.69[1.58-4.58]

4.18[1.97-8.90] 

1.00

2.51[1.48-4.25]**

2.67[1.56-4.56]***

4.13[1.94-8.80]***

Births in the last three years

No birth 

One birth

2 and above births 

1.00

0.72[0.60-0.87]

0.47[0.33-0.67] 

1.00

0.71[0.59-0.86] ***

0.53[0.37-0.75] ***

Age of respondent at first birth

Less than 18 years

18 – 24 

25 and above 

1.00

1.24[1.04-1.47]

0.96[0.70-1.30] 

1.00

1.28[1.08-1.52]

0.93[0.68-1.26]

Number of household members

1-4

5-8

9 and above 

1.00

0.95[0.79-1.14]

0.86[0.63-1.18] 

1.00

0.94[0.79-1.13]

0.90[0.66-1.24]

Number of under five children in household

None 

1-2

3 and above 

1.00

1.79[1.45-2.22]

0.90[0.60-1.36] 

1.00

1.80[1.46-2.22] ***

1.02[0.68-1.54]

Religion 

Muslim 

Orthodox 

Protestant 

Catholic 

Other 

1.00

1.75[1.40-2.20] 

1.41[1.07-1.85]

1.80[0.81-4.01]

0.76[0.36-1.59] 

1.00

1.86[1.46-2.36] ***

1.37[1.02-1.83]

2.12[0.91-4.92]

0.92[0.42-1.98]

Region 

Tigray 

Afar 

Amhara 

Oromia 

Somali 

Benishangul 

SNNPR

Gambela 

Harari 

Addis Ababa

Dire Dawa 

1.00

0.32[0.17-0.59] 

1.82[1.13-2.93]

1.73[1.05-2.84]

0.07[0.03-0.16] 

1.57[0.93-2.66]

2.02[1.21-3.35]

1.10[0.63-1.90]

1.11[0.64-1.95]

1.48[0.83-2.66]

0.87[0.49-1.53] 

1.00

0.41[0.22-0.78]**

2.33[1.43-3.80]**

2.06[1.24-3.43]**

0.12[0.05-0.27] ***

1.86[1.09-3.18]*

2.35[1.40-3.94] **

1.07[0.61-1.88]

1.11[0.62-1.96]

1.68[0.93-3.06]

1.06[0.59-1.88]

Type of place of residence 

Urban 

Rural 

1.00

0.77[0.40-0.72] 

1.00

0.77[0.55-1.06]

*P<0.05, **P<0.001, ***P<0.0001

Lines 278-280

In this study, household wealth had effects on modern contraceptive method utilization where the odds of modern contraceptive utilization among married women from households with middle and high income status were higher than women from households with low income status.

Reviewer #2: 

Comment: 

Please thoroughly revise the manuscript for language and grammar correction. 

Response:

We have tried to correct language and grammar errors together with the language expert.

 

Four out of ten married women utilized modern contraceptive method in Ethiopia: A Multilevel analysis of the 2019 Ethiopia mini demographic and health survey 

Sewunet Sako Shagaro1*, Teshale Fikadu Gebabo2 and Be’emnet Tekabe Mulgeta1

1Department of Health Informatics, School of Public Health, College of Medicine and Health Sciences, Arba Minch University, Arba Minch, Ethiopia

2Department of Public Health, School of Public Health, College of Medicine and Health Sciences, Arba Minch University, Arba Minch, Ethiopia 

* zesew1lalem@gmail.com (SS) 

Abstract 

Background: Modern contraceptive method is a product or medical procedure that interferes with reproduction from acts of sexual intercourse. Globally in 2019, 44% of women of reproductive age were using a modern method of contraception but it was 29% in sub-Saharan Africa. Therefore, the main aim of this analysis was to assess the prevalence of modern contraceptive utilization and associated factors among married women in Ethiopia.

Method: The current study used the 2019 Ethiopia mini demographic and health survey dataset. Both descriptive and multilevel mixed-effect logistic regression analysis were done using STATA version 14. A p-value of less than 0.05 and an adjusted odds ratio with a 95% confidence interval were used to report statistically significant factors with modern contraceptive utilization.

Result: The overall modern contraceptive utilization among married women in Ethiopia was 38.7% (95% CI: 37.3% to 40.0%). Among the modern contraceptive methods, injectables were the most widely utilized modern contraceptive method (22.82%) followed by implants (9.65%) and pills (2.71%). Maternal age, educational level, wealth index, number of living children, number of births in the last three years, number of under 5 children in the household, religion, and geographic region were independent predictors of modern contraceptive utilization. 

Conclusion: In the current study only four out of ten married non-pregnant women of reproductive age utilized modern contraceptive methods. Furthermore, the study has identified both individual and community-level factors that can affect the utilization of modern contraceptive methods by married women in the country. Therefore, concerned bodies need to improve access to reproductive health services, empower women through community-based approaches, and minimize region wise discrepancy to optimize the utilization.

Keywords: modern contraceptive method, reproductive age, Ethiopia 

Introduction 

Family planning is defined as the choice of individuals or couples to anticipate and attain their desired number of children, and the spacing and timing of their births. This can be realized through use of contraceptive methods and the treatment of involuntary infertility(1). Ensuring universal access to sexual and reproductive health-care services, including for family planning, information and education, and the integration of reproductive health into national strategies and programmes; and reducing the maternal mortality ratio to less than 70 per 100,000 live births by 2030 are some of goals directly targeted to family planning mentioned under sustainable development goal (SDG) 3 of United Nations (UN)(2). 

Modern contraceptive method is a product or medical procedure that interferes with reproduction from acts of sexual intercourse(3). Modern methods of contraceptive include oral contraceptives, sterilization (male and female), interuterine contraceptive device (IUD), injectables, implants, condoms (male and female), diaphragm, lactational amenorrhea method, standard days method (SDM), emergency contraception, cervical caps, contraceptive sponges and other country specific modern methods(4). 

Literatures have shown that contraceptive utilization can save the loss of more than 3 million girls per year due to unsafe abortion. It also improves school dropout, poverty, high rate of teenage marriage and complications related to unwanted pregnancy. It is believed that access to family planning methods gives women more freedom, independence and gender equity. Moreover, contraceptives utilization plays a significant role in improving maternal and child health, women empowerment, educational advances and economic development of the country. It allows the couples to decide on the desired number of children and to space births according to their plan and income level(5–8). Therefore, realizing the rights and well-being of women can be assured by improving access to high quality, affordable sexual and reproductive health services and information mainly about the full range of family planning methods(9). 

As indicated in the World Fertility and Family Planning 2020, globally 44% of women of reproductive age were utilizing modern contraceptive methods in 2019. The use of contraception among women of child bearing age in Sub-Saharan Africa increased from 13% in 1990 to 29% in 2019(10). The report also stated that the decrease of fertility in Sub-Saharan Africa has been quite slow as compared to other regions of the world. Consequently, Sub-Saharan Africa requires 34 years (from 1995 to 2029) to decline its fertility from 6.0 to 4.0 live births per woman while it took 24 years in Eastern and South-Eastern Asia for the same reduction. Also studies conducted in various countries of Africa revealed the prevalence of modern contraceptive methods utilization among married women continues to be low(11–15). 

However, there was progress in utilization of contraceptives among women of reproductive age in the East Africa region on average from 21% (1990-94) to 39% (2010-14). The highest proportion of contraceptive users was reported from Kenya (53%) while the lowest users were from Uganda by 23% from the region(16,17). In east and other parts of Africa, injectables were the most widely utilized modern contraceptive method followed by implants(16,18–21). 

Various studies have identified maternal age, educational status, wealth index, births in the last three years, marital status, religion, region, and place of residence as individual-level and community-level factors associated with modern contraceptive utilization in Ethiopia and Africa as well(11,15,22,23). Other studies also considered husband’s education level, media exposure, number of desired children and fertility related decisions as factors that may influence contraceptive utilization(13,21,24,25). As stated in literature, higher educational status of women was positively associated with modern contraceptive utilization where the odds of modern contraceptive utilization among women who attained primary educational levels was higher than women without formal education(12–15,17,26,27). Similarly, modern contraceptive utilization was highly influenced by the wealth index of the women(11,14,15,25). 

Despite the reduction of the total fertility rate over the years, Ethiopia has still one of the highest fertility rates in Africa. The prevalence of any modern contraceptive use by currently married Ethiopian women has tripled over the last fifteen years, jumping from 14% in 2005 to 41% in 2019(28). The most commonly used contraceptive methods for currently married women in the country are injectables (27%), followed by implants (9%). The lowest contraceptive utilization was reported from the Somali region (3%) and the highest from both Amhara and Oromia regions (50%) of Ethiopia(28). However, as stated in the 2020 report of the Federal Ministry of Health of Ethiopia, the health sector transformation plan (HSTP) I target (55%) of the national contraceptive prevalence rate was not achieved. And all the regions in the country were unable to attain their target set for the year(29). These evidence clearly convey that the modern contraceptive method utilization remains a great public health problem in Ethiopia. Therefore, this study aimed to assess the prevalence and associated factors of modern contraceptive utilization among married women in Ethiopia using the 2019 Ethiopia mini demographic and health survey data. 

Methods

Study settings and data source

The study used the 2019 Ethiopia mini demographic and health survey (EMDHS) dataset. It is the second mini demographic and health survey which was conducted in March, 2019, to June, 2019 in Ethiopia. Ethiopia is divided into two administrative cities and nine regions. All married women in the reproductive age living in nine regions and two administrative cities of Ethiopia were included in the study. 

Sampling procedures

The EMDHS used a complete list of 149,093 enumeration areas (EAs) created for the upcoming Ethiopia population and housing census as a sampling frame. The frame comprises information about the EA location, type of residence (urban or rural), and estimated number of residential households. Accordingly, the sample was stratified and selected in two-stages(28). 

Out of the total EAs, a total of 305 EAs (93 in urban areas and 212 in rural areas) were selected with probability proportional to EA size and with independent selection in each sampling stratum for the survey. Secondly, a fixed number of 30 households per cluster were selected with an equal probability systematic selection from the newly created household listing. 

Finally, the 2019 EMDHS survey covered 8,663 households out of the selected 8,794 households providing a response rate of 99%. About 8,885 women completed the interview from 16,583 women identified for the interview, yielding a response rate of 99%. According to the EMDHS report response rates were higher in rural than in urban areas(28). 

The source population of this study was all married non-pregnant women who were in the reproductive age group and living in Ethiopia. Pregnant mothers and those who were not in union at the time of survey were excluded from the study. Hence, 4,983 married women data were extracted from the 2019 EMDHS datasets. 

Measurements of variables

Dependent variable: Modern contraceptive utilization was the outcome variable of the study. Woman was considered as a “utilizer” if she had been utilizing any modern contraceptive methods such as oral contraceptives, male and female sterilization, intrauterine contraceptive device, injectables, implants, male and female condoms, lactational amenorrhea method, standard days method, and emergency contraception(4,30) during the 2019 EMDHS survey period while woman who had been utilizing traditional, folkloric or no method was considered as a “non-utilizer”. 

Independent Variable: Both the individual and community level explanatory variables were used to assess modern contraceptives utilized among women in childbearing age in the country.

Individual level factors: maternal age, educational level of mother, wealth index, total children ever born, number of living children, births in the last three years, age of respondent at first birth, family size, number of under five children in household, and knowledge on modern contraceptive method. 

Community-level variables: community level explanatory variables include religion, region, and place of residence (Table 1). 

Table 1: Description of individual and community level variables 

Variables Description

Individual-level variables 

Maternal age It is the current age of women recoded as 15-19, 20-24, 25-29, 30-34, 35-39, 40-44, and 45-49. 

Educational level of mother This is the level of education a woman attained and recoded as no education, primary, secondary and higher. 

Wealth index In DHS the wealth index is calculated using data on a household’s ownership of selected assets. Each household asset is assigned a weight score generated through PCA. The resulting asset scores are standardized and summed by household and individuals are ranked according to the total score of the household in which they reside. Finally, it is grouped as poorest, poorer, middle, richer and richest. It is recoded as Low Income (poorest and poorer), Middle Income and High Income (richer and richest).

Total children ever born It is recoded as 1-3 children, 4-6 children, and 7 and more

Number of living children Number of living children is recoded as no child, 1-4, 5-8, 9 and above children.

Births in the last three years This variable is recoded as no birth, one birth, and 2 and above births 

Age of respondent at 1st birth Recoded as less than 18 years, 18-24, and 25 and above years 

Family size It is number of household members and recoded as 1-4, 5-8, and 9 and above

Number of <5 children in household Number of children 5 and under in household is recoded as none, 1-2, and 3 and above children.

Knowledge on modern contraceptive method Knowledge of contraceptive method is defined as percentage who know of any method, any modern method, any traditional method and specific methods; mean number of methods known. It is recoded as “yes” if the women say they know or have heard at least one of the modern contraceptive methods and else recoded as “no”. 

Religion This variable is the religious group to which the woman associates herself and recoded as Muslim, Orthodox, Protestant, Catholic and other.

Region Region of residence is typically the first administrative level within the country, or a grouping of the first administrative level. It is grouped as Tigray, Afar, Amhara, Oromia, Somali, Benishangul, SNNPR, Gambela, Harari, Addis Ababa and Dire Dawa.

Place of residence It is the designation of the cluster or enumeration area as an urban area or a rural area.

PCA: principal components analysis

Data processing and statistical analysis

The data were extracted, cleaned, re-coded, and analyzed using STATA version 14. Descriptive statistics were presented using graphs, tables and narrations. A multilevel mixed-effect logistic regression analysis, an advanced model, was used to overcome the violation of independence of observations and equal variance assumption of the traditional logistic regression model due to a hierarchical nature of DHS data. 

We first estimated an intercept-only model or the null model (with only the outcome variable). Secondly, we included all individual level factors in the model (model I). In the third stage, we constructed model II (fitted with community-level factors only) and finally model III was fitted with both individual and community-level factors. To examine clustering and the extent to which community-level factors explain the unexplained variance of the null model, the intraclass correlation coefficient (ICC) and a proportional change in variance (PCV) were checked. The model with the lowest deviance, model III, was selected as the best-fitted model for the analysis. Variance inflation factor (VIF) was done to test the existence of multicollinearity among the independent variables.

Variables having a p-value of less than 0.2 in the bivariable analysis were selected as candidate variables for the multivariable mixed-effect logistic regression analysis. In the final model, a p-value of less than 0.05 and an adjusted odds ratio (AOR) with a 95% confidence interval (CI) were used to report statistically significant factors with modern contraceptive utilization among childbearing age women.

Ethical consideration 

This study used secondary data from demographic and health survey repositories. The survey data do not contain all identifying information. The MEASURE Demographic and Health Survey Program team allowed us to access the data upon sending an abstract of our study to an online request form http://www.measuredhsprogram.com.

Results

Socio-demographic characteristics of the study participants 

A total of 4,983 non-pregnant married women were included in the analysis. The mean age of respondents was 30.7 years (SD ± 8.29 years). From the total number of respondents, almost one quarter (23.48%) were in the age range of 25-29 years. More than half (52.26%) of the women included in the analysis had no education whereas only 314(6.30%) respondents attended higher education. In this study, women who were from low income households (41.26%) were nearly equal to women who were from high income households (42.83%). 

More than one quarter (26.73%) of the respondents had 1-3 children ever born and about 2965 (59.5%) of the women have had 1-4 number of live children. One half of the respondents (49.75%) had no birth while 352 (7.06 %) respondents had two and more births in the last three years. Out of the total respondents, one half (50.35%) were in the age range of less than 18 years whereas 842(16.9%) were in the age range of 25 and above years during their first birth. Regarding family size, 2,652(53.22%) households have had 5-8 family members and 3,088(61.97%) households have had 1 or 2 child/children aged less than 5 years. Majority of the women (93.62%) included in the survey know modern contraceptives methods.

Concerning the community-level factors, the majority (42.24%) were followers of Muslim religion and nearly three fourth (72.61%) of the women were rural residents. Slightly higher proportion of women (12.88%) were from Oromia region and the lowest proportion (6.64%) were from Addis Ababa city administration (Table 2).

Table 2: Distribution of individual and community-level factors of the married women in Ethiopia, EMDHS 2019

Individual variables Number (N) Percent (%)

Maternal age

15-19

20-24

25-29

30-34

35-39

40-44

45-49 

382

803

1170

870

810

546

402 

7.67

16.11

23.48

17.46

16.26

10.96

8.07

Educational level of mother

No education 

Primary 

Secondary 

Higher 

2604

1589

476

314 

52.26

31.89

9.55

6.30

Wealth index

Low Income

Middle Income

High Income 

2056

793

2134 

41.26

15.91

42.83

Total children ever born

1-3 children 

4-6 children 

7 and above 

2189

1462

1332 

43.93

29.34

26.73

Number of living children

No child

1-4 children

5-8 children

9 and above 

514

2965

1391

113 

10.32

59.50

27.91

2.27

Births in the last three years

No birth 

One birth

2 and above births 

2479

2152

352 

49.75

43.19

7.06

Age of respondent at first birth

Less than 18 years

18 – 24 

25 and above 

2509

1632

842 

50.35

32.75

16.90

Number of household members

1-4

5-8

9 and above 

1803

2652

528 

36.18

53.22

10.60

Number of under five children in household

None 

1-2

3 and above 

1521

3088

374 

30.52

61.97

7.51

Knowledge of any method

Knows no method

Knows only folkloric method 

Knows only traditional method

Knows modern method 

311

1

6

4665 

6.24

0.02

0.12

93.62

Religion 

Muslim 

Orthodox 

Protestant 

Catholic 

Other 

2,105

1,806

983

38

51 

42.24

36.24

19.73

0.76

1.02

Region 

Tigray 

Afar 

Amhara 

Oromia 

Somali 

Benishangul 

SNNPR

Gambela 

Harari 

Addis Ababa

Dire Dawa 

391

407

584

642

353

467

629

406

378

331

395 

7.85

8.17

11.72

12.88

7.08

9.37

12.62

8.15

7.59

6.64

7.93

Type of place of residence 

Urban 

Rural 

1365

3618 

27.39

72.61

Prevalence of modern contraceptive utilization

From the total women included in the analysis, 1,927 (38.7%) respondents utilized modern contraceptive methods. In this study, non-modern contraceptive users were those who utilized traditional methods, folkloric methods or no method at the time of the survey. Accordingly, a large number of women did not utilize any contraceptive method while the least commonly utilized method was folkloric method (Fig. 1). Furthermore, injectables were the most widely utilized modern contraceptive method (22.82%) followed by implants (9.65%), and pills (2.71%). (Fig 2). 

Fig. 1: Percentage of current use by contraceptive method type among married women in Ethiopia, EMDHS 2019 

Fig. 2: Proportion of current use by contraceptive method among married women in Ethiopia, EMDHS 2019 

Factors affecting modern contraceptive utilization 

From all independent individual and community level factors, maternal age, educational level of women, wealth index, total children ever born, births in the last three years, age of mother at first birth, number of household members, number of under five children in household, religion, region and place of residence were eligible variables for multivariable multilevel analysis.

In the multivariable multilevel analysis, the following individual-level and community-level factors were associated with modern contraceptive utilization in Ethiopia. The odds of modern contraceptive utilization was 47% [AOR=0.53; 95%CI: 0.36-0.79], 67% [AOR=0.33; 95%CI: 0.21-0.51], and 87% [AOR=0.13; 95%CI: 0.08-0.22] lower among women who aged 35-39, 40-44, and 45-49 years as compared to women aged 15-19 years. A woman who attended primary, secondary, and higher school was 1.73[AOR=1.73; 95%CI: 1.45-2.06], 1.77[AOR=1.77; 95%CI: 1.34-2.33] and 2.56[AOR=2.56; 95%CI: 1.83-3.59] times more likely to utilize modern contraceptive method as compared to woman who had not attended formal education. The odds of modern contraceptive utilization among women from households with middle and high income status were 1.72[AOR=1.72; 95%CI: 1.38-2.15] and 1.88[AOR=1.88; 95%CI: 1.51-2.34] times higher as compared to those from households with low income status. Those women who had 1-4, 5-8, and 9 and above living children were 2.51[AOR=2.51; 95%CI: 1.48-4.25], 2.67[AOR=2.67; 95%CI: 1.56-4.56] and 4.13[AOR=4.13; 95%CI: 1.94-8.80] times more likely utilize modern contraceptive method as compared to those women who had no living child. Woman who had one birth, and two and above births in the last three years was 29% [AOR=0.71; 95%CI: 0.59-0.86] and 47% [AOR=0.53; 95%CI: 0.37-0.75] lowers odds of modern contraceptive utilization as compared to those woman who had no birth in the last three years. The odds of modern contraceptive utilization was 1.8[AOR=1.80; 95%CI: 1.46-2.22] times higher among women from households where the number of under five children were 1-2 as compared to women from households where there were no under five children. 

Among community level factors, the odds of modern contraceptive method utilization was 1.86[AOR=1.86; 95%CI: 1.46-2.36] and 1.37[AOR=1.37; 95%CI: 1.02-1.83] times higher among women who have been following Orthodox religion and Protestant religion as compared to women who have been following Muslim religion. The odds of modern contraceptive method utilization among women living in Amhara, Oromia, Benishangul and Southern nation nationalities and peoples regions of Ethiopia were 2.33[AOR=2.33; 95%CI: 1.43-3.80], 2.06[AOR=2.06; 95%CI: 1.24-3.43], 1.86[AOR=1.86; 95%CI: 1.09-3.18], and 2.35[AOR=2.35; 95%CI: 1.40-3.94] times higher as compared to women living in Tigray region, respectively. The odds of modern contraceptive method utilization among women living in Afar and Somali regions were lower by 59% [AOR=0.41; 95%CI: 0.22-0.78] and 88% [AOR=0.12; 95%CI: 0.05-0.27] as compared to women living in Tigray region of Ethiopia (Table 3). 

Table 3: Multivariable multilevel analysis for factors associated with modern contraceptive utilization among married women in Ethiopia, EMDHS 2019 

Variables Null Model Model I AOR(95% CI) Model II 

AOR(95%CI) Model III AOR(95%CI)

Maternal age

15-19

20-24

25-29

30-34

35-39

40-44

45-49 

1.00

1.27[0.92-1.74]

1.01[0.73-1.41]

0.86[0.60-1.24]

0.61[0.41-0.91]

0.39[0.25-0.60]

0.15[0.09-0.25] 

1.00

1.23[0.89-1.69]

0.92[0.67-1.28]

0.77[0.54-1.11]

0.53[0.36-0.79]**

0.33[0.21-0.51]***

0.13[0.08-0.22]***

Educational level of mother

No education 

Primary 

Secondary 

Higher 

1.00

1.73[1.45-2.06] 

1.77[1.34-2.33]

2.56[1.83-3.59] 

1.00

1.62[1.35-1.93] ***

1.50[1.14-1.98]**

2.06[1.47-2.88] ***

Wealth index

Low Income

Middle Income

High Income 

1.00

1.72[1.38-2.15]

1.88[1.51-2.34] 

1.00

1.50[1.21-1.86]***

1.63[1.30-2.05]***

Total children ever born

1-3 children 

4-6 children 

7 and above 

1.00

1.13[0.90-1.42]

0.99[0.70-1.39] 

1.00

1.14[0.90-1.43]

1.01[0.72-1.42]

Number of living children

No child

1-4 children

5-8 children

9 & above 

1.00

2.57[1.52- 4.35]

2.69[1.58-4.58]

4.18[1.97-8.90] 

1.00

2.51[1.48-4.25]**

2.67[1.56-4.56]***

4.13[1.94-8.80]***

Births in the last three years

No birth 

One birth

2 and above births 

1.00

0.72[0.60-0.87]

0.47[0.33-0.67] 

1.00

0.71[0.59-0.86] ***

0.53[0.37-0.75] ***

Age of respondent at first birth

Less than 18 years

18 – 24 

25 and above 

1.00

1.24[1.04-1.47]

0.96[0.70-1.30] 

1.00

1.28[1.08-1.52]

0.93[0.68-1.26]

Number of household members

1-4

5-8

9 and above 

1.00

0.95[0.79-1.14]

0.86[0.63-1.18] 

1.00

0.94[0.79-1.13]

0.90[0.66-1.24]

Number of under five children in household

None 

1-2

3 and above 

1.00

1.79[1.45-2.22]

0.90[0.60-1.36] 

1.00

1.80[1.46-2.22] ***

1.02[0.68-1.54]

Religion 

Muslim 

Orthodox 

Protestant 

Catholic 

Other 

1.00

1.75[1.40-2.20] 

1.41[1.07-1.85]

1.80[0.81-4.01]

0.76[0.36-1.59] 

1.00

1.86[1.46-2.36] ***

1.37[1.02-1.83]

2.12[0.91-4.92]

0.92[0.42-1.98]

Region 

Tigray 

Afar 

Amhara 

Oromia 

Somali 

Benishangul 

SNNPR

Gambela 

Harari 

Addis Ababa

Dire Dawa 

1.00

0.32[0.17-0.59] 

1.82[1.13-2.93]

1.73[1.05-2.84]

0.07[0.03-0.16] 

1.57[0.93-2.66]

2.02[1.21-3.35]

1.10[0.63-1.90]

1.11[0.64-1.95]

1.48[0.83-2.66]

0.87[0.49-1.53] 

1.00

0.41[0.22-0.78]**

2.33[1.43-3.80]**

2.06[1.24-3.43]**

0.12[0.05-0.27] ***

1.86[1.09-3.18]*

2.35[1.40-3.94] **

1.07[0.61-1.88]

1.11[0.62-1.96]

1.68[0.93-3.06]

1.06[0.59-1.88]

Type of place of residence 

Urban 

Rural 

1.00

0.77[0.40-0.72] 

1.00

0.77[0.55-1.06]

*P<0.05, **P<0.001, ***P<0.0001

Regarding model fitness, as depicted in Table 4, intra-cluster correlation coefficient (ICC) of 28.97% (in the null model) confirmed the choice of multilevel model for the analysis. It indicates that 28.97% of the variation in modern contraceptive utilization among married women was due to the variation between clusters. Also the highest PCV (93.70%) in the last model (Model III) show most of the variations of modern contraceptive utilization were attributable to both individual and community level factors. Furthermore, the lowest deviance in the final model confirmed that the model (Model III) was the best fitted-model (Table 4). 

Table 4: Random effect analysis and model fitness for assessing factors associated with modern contraceptive utilization among married women in Ethiopia, EMDHS 2019

Parameters Null Model Model I Model II Model III

Community level variance (SE) 0.16 0.1458 0.0819 0.0864

ICC 0.2897 0.2453 0.1441 0.1411

PCV Reference 54.89% 92.95% 93.70

Model fitness

Log-likelihood -3039.41 -2775.74 -2927.37 -2683.76

Deviance 6078.82 5551.48 5854.74 5367.52

AIC 6082.82 5603.48 5888.75 5449.52

BIC 6095.85 5772.85 5999.48 5716.58

Discussion

 The prevalence of modern contraceptive utilization among married women in Ethiopia was 38.7% (95% CI: 37.3% to 40.0%). This finding is higher than the result of the secondary data analysis of the 2016 Ethiopia demographic and health survey where the prevalence of modern contraceptive utilization was 20.42%(11). It was also higher than the studies conducted in different parts of Ethiopia like Bale zone (20.8%), Metekel zone(18.6%), and Afar region(8.5%); and other African countries such as Ghana(21%), Burkina Faso(24%), Senegal(26.3%) and Nigeria(10.3%) (13–15,22,26,31,32). The current prevalence was higher than the national figure(35%) according to the 2016 EDHS report and the findings of the secondary data analysis of the 2016 EDHS (34.9%)(23,30). However, it was similar to a study done in Northwest Ethiopia, which was 37%(33). On the other hand, the utilization rate was lower when compared to the study conducted in Western Ethiopia (71.9%)(24), Southern Ethiopia (69.5%, 53.3%)(19,34), Northern Ethiopia (51.3%, 66.2%)(18,25), Yaoundé-Cameroon(58.9%)(35), and India(48.6%)(36). These differences could be due to participants’ cultural, religious and awareness differences on modern contraceptive methods. Additionally, variation of the study period might contributed for the observed discrepancy of modern contraceptive methods utilization.

As evidenced by the literature, the prevalence of unplanned pregnancy is lower among younger women than older women(37). This fact is supported by our study finding where the odds of modern contraceptive utilization was lower among aged women as compared to young women. This finding is also in agreement with the study findings done in Amhara regional state of Ethiopia(25), Gondar(38), East Africa(17) and findings from secondary data analysis of the 2016 EDHS(23). The possible reason might be related with the woman’s educational status. In our study, aged women were less educated as compared to the young women. Previous studies have reported that as maternal educational level increase the likelihood of desire for bearing more children decreases(39). Also there are evidences that show aged women may be in less need of modern contraceptive methods as they prefer traditional methods than the young women(40). 

Our finding also revealed that the educational level of the mother was an independent predictor of modern contraceptive method utilization among married women in Ethiopia. Women who attended primary, secondary and higher education were more likely to utilize modern contraceptive methods as compared to women with no education. This finding is in line with studies conducted in Western Ethiopia(24), Northwest Ethiopia(26,32), Ghana(12,41) and Nigeria(14). This might be due to an increase in the level of education, enhances awareness on sexual and reproductive health, maximizes probability of getting contraceptive related information and improves decision making ability of women. 

In this study, household wealth had effects on modern contraceptive method utilization where the odds of modern contraceptive utilization among married women from households with middle and high income status were higher than women from households with low income status. This finding was concordant with the findings from secondary data analysis of Ethiopia demographic and health survey(25,30), three demographic and health surveys in Bangladesh(42), a community based cross-sectional study done in Western Ethiopia(24), Eastern Ethiopia(32) and Nigeria (14). In Ethiopia, all public health facilities provide modern contraceptives for any woman in the child bearing age (15-49 years) free of charge, hence household income status has no effect in obtaining the service. However, a woman household’s wealth has great influence on exposure to education, access to basic health services and health information.

In this study, the odds of modern contraceptive utilization among women who had one or more children were higher as compared to women who had no child. This finding is supported by the study finding conducted in Southern nation nationalities and peoples region(19), North Shoa zone(43) and Senegal(15). This could be due to women who had one or more children might have greater need to space or limit births than those women who had no child. This evidence is also supported by the current study which reported that a woman from a household with one or two under five children was more likely to utilize modern contraceptive method as compared to a woman who had no child aged less than five.

This study revealed that births in the last three years were significantly associated with modern contraceptive utilization. Women who had one birth or who had two and more births were 33% and 47% times less likely to utilize modern contraceptive methods as compared to women who had no birth in the last three years respectively. The finding is consistent with the result of the study conducted in Ethiopia where modern contraceptive utilization was lower among women who had one birth or two and more births as compared to women who had no birth in the last three years (23) (43). The possible reason for this could be the variation in their desire to have more children. The problem of reverse causality might be also another reason as most of the women who reported as present utilizers may started utilizing the methods two or three years ago. 

Multivariate multilevel analysis also showed that religion of the woman was independently associated with modern contraceptive utilization. Women who had been following Orthodox religion and Protestant religion had higher odds of utilizing modern contraceptives as compared to those women who had been following Muslim religion. This finding was consistent with secondary data analysis of three demographic and health survey in Bangladesh where Muslim women have lower odds of contraceptive utilization compared with non-Muslim women(42,44). This finding was also in agreement with secondary data analysis of Ethiopia demographic and health survey in 2016 (23). 

Women’s region of residence was also associated with modern contraceptive method utilization among married women in Ethiopia. Women who live in Afar and Somali regions of Ethiopia were less likely to utilize modern contraceptive methods as compared to women who live in Tigray region. In contrast, women who reside in Amhara, Oromia, Benishangul and SNNPR regions of Ethiopia were more likely to utilize modern contraceptive methods as compared to women who live in Tigray region. The finding was in line with secondary data analysis of the 2016 Ethiopia demographic and health survey where a woman who resides in the Afar and Somali regions was nearly 55% and 93% times less likely to utilize modern contraceptive as compared to a woman who resides in Addis Ababa(11). 

The strength of the current study includes use of multilevel mixed effect analysis to overcome the hierarchical nature of EMDHS data, and deploying nationally weighted representative and most recent EMDHS data which shows country level proportion of contraceptive utilization and its associated factors among women in child bearing age. The study also has some limitations. Showing temporal relationship between modern contraceptives utilization and its predictors was impossible due to the type of the study design, cross-sectional, used for the survey. Moreover, the EMDHS data did not encompass information about some predictors of modern contraceptive utilization, as it was a mini report. Conversely, the investigators have trust that the stated limitations cannot impose significant impact on the validity of study findings. 

Conclusion 

In the current study only four out of ten married non-pregnant women of reproductive age utilized modern contraceptive methods. Furthermore, the study has identified both individual and community-level factors that can affect the utilization of modern contraceptive methods by married women in the country. Therefore, concerned bodies need to improve access to reproductive health services, empower women through community-based approaches, and minimize region wise discrepancy to optimize the utilization.

Acknowledgments

We would like to forward our kindest regards to MEASURE Demographic and Health Survey Program for providing us 2019 EMDHS data. 

Authors’ contribution

SSS designed the study, requested data, processed and analyzed data, and drafted the manuscript. TFG and BTM were equally involved in the processing, analysis, and drafting the manuscript. All authors have read and approved the final manuscript.

Abbreviations

AOR: Adjusted Odds Ration, CI: Confidence Interval, CSA: Central Statistical Agency, EMDHS: Ethiopia Mini Demographic and Health Survey, HSTP: Health Sector Transformation Plan, IUD: Interuterine Contraceptive Device, SDM: Standard Days Method 

Funding 

Not applicable

Availability of data and materials

The dataset used for the current study is available in the demographic and health survey repository in https//www.dhsprogram.com/data. 

Consent to publication

Not applicable

Competing interests

The authors declare that they haves no competing interests.

References

1. Edition T, Ababa A. Federal Democratic Republic of Ethiopia Ministry of Health Third Edition National Guideline for Family Planning Services In Ethiopia Third Edition. 2020;(July):1–77. 

2. Inter-Agency and Expert Group on Sustainable Development Goal Indicators. Report of the Inter-Agency and Expert Group on Sustainable Development Goal Indicators (E/CN.3/2016/2/Rev.1), Annex IV. Rep Inter-Agency Expert Gr Sustain Dev Goal Indic [Internet]. 2016;Annex IV. Available from: https://sustainabledevelopment.un.org/content/documents/11803Official-List-of-Proposed-SDG-Indicators.pdf

3. Hubacher D, Trussell J. A definition of modern contraceptive methods. Contraception [Internet]. 2015;92(5):420–1. Available from: http://dx.doi.org/10.1016/j.contraception.2015.08.008

4. Croft, Trevor N., Aileen M. J. Marshall, Courtney K. Allen et al. Guide to DHS Statistics. Rockville, Maryland, USA ICF. 2018;22–51. 

5. Region M. Health Gender. 2020;4–5. 

6. WHO. Adolescent pregnancy fact sheet. 2014. 

7. Schaapveld, A & Ineke V. Contraceptives Knowledge among High School Female Adolescents. Reprod Health. 2018;2(1). 

8. Kassa GM, Arowojolu AO, Odukogbe AA, Yalew AW. Prevalence and determinants of adolescent pregnancy in Africa: a systematic review and Meta-analysis. Reprod Health. 2018;15(1):1–17. 

9. Global FP. 2018 EDITION What ’ s New in This Edition ? 2018. 

10. World Fertility and Family Planning 2020: Highlights. World Fertility and Family Planning 2020: Highlights. 2020. 

11. Gebre MN, Edossa ZK. Modern contraceptive utilization and associated factors among reproductive-age women in Ethiopia: Evidence from 2016 Ethiopia demographic and health survey. BMC Womens Health. 2020;20(1):1–14. 

12. Apanga PA, Adam MA. Factors influencing the uptake of family planning services in the Talensi district, Ghana. Pan Afr Med J. 2015;20:1–9. 

13. O’Regan A, Thompson G. Indicators of young women’s modern contraceptive use in Burkina Faso and Mali from Demographic and Health Survey data. Contracept Reprod Med. 2017;2(1):1–8. 

14. Ofonime JE. Determinants of modern contraceptive uptake among Nigerian women: Evidence from the national demographic and health survey. Afr J Reprod Health [Internet]. 2017;21(3):89–95. Available from: https://www.scopus.com/inward/record.uri?eid=2-s2.0-85038842247&partnerID=40&md5=c36f097c56e5413d445dcaaf7d7a30a1

15. Zegeye B, Ahinkorah BO, Idriss-Wheeler D, Olorunsaiye CZ, Adjei NK, Yaya S. Modern contraceptive utilization and its associated factors among married women in Senegal: a multilevel analysis. BMC Public Health. 2021;21(1):1–13. 

16. Sheet F. Contraceptive use in East Africa: What do the numbers tell us? 2020;(October 2018):1–4. 

17. Tessema ZT, Teshale AB, Tesema GA, Yeshaw Y, Worku MG. Pooled prevalence and determinants of modern contraceptive utilization in East Africa: A Multi-country Analysis of recent Demographic and Health Surveys. PLoS One [Internet]. 2021;16(3 March):1–16. Available from: http://dx.doi.org/10.1371/journal.pone.0247992

18. TB Kassa GDZB. Assessment of modern contraceptive practice and associated factors among currently married women age 15-49 years in Farta District, South Gondar zone. North west Ethiop. 2015;2(6):507–12. 

19. Endriyas M, Eshete A, Mekonnen E, Misganaw T, Shiferaw M, Ayele S. Contraceptive utilization and associated factors among women of reproductive age group in Southern Nations Nationalities and Peoples’ Region, Ethiopia: cross-sectional survey, mixed-methods. Contracept Reprod Med [Internet]. 2017;2(1):1–9. Available from: http://dx.doi.org/10.1186/s40834-016-0036-z

20. Alemayehu GA, Fekadu A, Yitayal M, Kebede Y, Abebe SM, Ayele TA, et al. Prevalence and determinants of contraceptive utilization among married women at Dabat Health and Demographic Surveillance System site, northwest Ethiopia. BMC Womens Health. 2018;18(1):1–7. 

21. Debebe S, Limenih MA, Biadgo B. Modern contraceptive methods utilization and associated factors among reproductive aged women in rural Dembia District, northwest Ethiopia: Community based cross-sectional study. 2020;15(6):367–74. 

22. Beson P, Appiah R, Adomah-Afari A. Modern contraceptive use among reproductive-aged women in Ghana: Prevalence, predictors, and policy implications. BMC Womens Health. 2018;18(1):1–8. 

23. Abate MG, Tareke AA. Individual and community level associates of contraceptive use in Ethiopia: A multilevel mixed effects analysis. Arch Public Heal. 2019;77(1):1–12. 

24. Tekelab T, Melka AS, Wirtu D. Predictors of modern contraceptive methods use among married women of reproductive age groups in Western Ethiopia: A community based cross-sectional study. BMC Womens Health [Internet]. 2015;15(1):1–8. Available from: http://dx.doi.org/10.1186/s12905-015-0208-z

25. Melash Belachew Asresie , Gedefaw Abeje Fekadu, Gizachew Work Dagnew YMG. Modern Contraceptive Use and Influencing Factors in Amhara Regional State: Further Analysis of Ethiopian Demographic Health Survey Data 2016. Hindawi Adv Public Heal. 2020:8 pages. 

26. Adane AA, Bekele YA, Melese E, Worku GT, Netsere HB. Modern Contraceptive Utilization and Associated Factors among Married Gumuz Women in Metekel Zone North West Ethiopia. Biomed Res Int. 2020;2020. 

27. Andualem Samuel, Abraham Uliso, Birke Olle, Desalech Dambe MN and MMS. Assessment of Modern Contraceptive Method Utilization and Associated Factors Among Women of Reproductive Age Group in Arba Minch Town, SNNPR, Ethiopia. EC Gynaecol. 2017;6.2:36–53. 

28. Ethiopian Public Health Institute (EPHI) and ICF. 2019. Mini Demographic and Health Survey 2019: key Indicators. Rockville, Maryland, USA: EPHI and ICF. 2019. 35 p. 

29. FMOH Performance report. Federal Ministry of Health of Ethiopia Performance report 2012 EFY (2019/2020). 2020;2012:487–567. 

30. Central Statistical Agency (CSA) [Ethiopia] and ICF. Ethiopia Demographic and Health Survey 2016. Addis Ababa, Ethiop Rockville, Maryland, USA CSA ICF. 

31. Belda SS, Haile MT, Melku AT, Tololu AK. Modern contraceptive utilization and associated factors among married pastoralist women in Bale eco-region, Bale Zone, South East Ethiopia. BMC Health Serv Res. 2017;17(1):1–12. 

32. Alemayehu M, Lemma H, Abrha K, Adama Y, Fisseha G, Yebyo H, et al. Family planning use and associated factors among pastoralist community of afar region, eastern Ethiopia. BMC Womens Health [Internet]. 2016;16(1):1–9. Available from: http://dx.doi.org/10.1186/s12905-016-0321-7

33. Geremew AB, Gelagay AA. Modern contraceptive use and associated factors among married women in Finote Selam town Northwest Ethiopia: a community based cross-sectional study. Women’s Midlife Heal. 2018;4(1):1–8. 

34. A E. Contraceptive Method Mix Utilization and its Associated Factors among Married Women in Gedeo Zone, Southern Nations, Nationality and People Region-Ethiopia: A Community based Cross Sectional Study. Epidemiol Open Access. 2015;05(04). 

35. Njotang PN, Yakum MN, Ajong AB, Essi MJ, Akoh EW, Mesumbe NE, et al. Determinants of modern contraceptive practice in Yaoundé-Cameroon: A community based cross sectional study. BMC Res Notes. 2017;10(1):4–9. 

36. Dey AK. Socio-demographic determinants and modern family planning usage pattern-an analysis of National Family Health Survey-IV data. Int J Community Med Public Heal. 2019;6(2):738. 

37. Tsegaye AT, Mengistu M, Shimeka A. Prevalence of unintended pregnancy and associated factors among married women in west Belessa Woreda, Northwest Ethiopia, 2016. Reprod Health. 2018;15(1):1–8. 

38. Oumer M. Modern Contraceptive Method Utilization and Associated Factors Among Women of Reproductive Age in Gondar City , Northwest Ethiopia. 2020;53–67. 

39. Akinyemi JO, Odimegwu CO. Social contexts of fertility desire among non-childbearing young men and women aged 15–24 years in Nigeria. Reprod Health [Internet]. 2021;18(1):1–18. Available from: https://doi.org/10.1186/s12978-021-01237-1

40. Godfrey EM, Chin NP, Fielding SL, Fiscella K, Dozier A. Contraceptive methods and use by women aged 35 and over: A qualitative study of perspectives. BMC Womens Health [Internet]. 2011;11(1):5. Available from: http://www.biomedcentral.com/1472-6874/11/5

41. Aviisah PA, Dery S, Atsu BK, Yawson A, Alotaibi RM, Rezk HR, et al. Modern contraceptive use among women of reproductive age in Ghana: Analysis of the 2003-2014 Ghana Demographic and Health Surveys. BMC Womens Health. 2018;18(1):1–10. 

42. Haq I, Sakib S, Talukder A. Sociodemographic Factors on Contraceptive Use among Ever-Married Women of Reproductive Age: Evidence from Three Demographic and Health Surveys in Bangladesh. Med Sci. 2017;5(4):31. 

43. A. M, D. W, A. F, B. M. Determinants of modern contraceptive utilization among married women of reproductive age group in North Shoa Zone, Amhara Region, Ethiopia. Reprod Health [Internet]. 2014;11(1):1–7. Available from: http://ovidsp.ovid.com/ovidweb.cgi?T=JS&PAGE=reference&D=emed12&NEWS=N&AN=2014111648

44. Kibria GMA, Hossen S, Barsha RAA, Sharmeen A, Paul SK, Uddin SMI. Factors affecting contraceptive use among married women of reproductive age in Bangladesh. J Mol Stud Med Res. 2016;2(1):70–9. 

Comment:

Use appropriate and more scientific words like 'reported' instead of 'declared'. In Introduction, line (32), use appropriate word like 'choice' instead of 'ability'.

Response:

We have substituted appropriate terms/words in place of declare and other similar terms

Lines 155-157

In the final model, a p-value of less than 0.05 and an adjusted odds ratio (AOR) with a 95% confidence interval (CI) were used to report statistically significant factors with modern contraceptive utilization among childbearing age women.

Line 36 and 37

Family planning is defined as the choice of individuals or couples to anticipate and attain their desired number of children, and the spacing and timing of their births.

Comment:

Clearly mention which methods are considered as 'Modern Contraceptive Methods'. 

Response:

Under the methods section (Measurement of variables), we have mentioned a list of modern contraceptive methods considered in this study based on the 2019 EMDHS report.

Lines 122-127

Dependent variable: Modern contraceptive utilization was the outcome variable of the study. Woman was considered as a “utilizer” if she had been utilizing any modern contraceptive methods such as oral contraceptives, male and female sterilization, intrauterine contraceptive device, injectables, implants, male and female condoms, lactational amenorrhea method, standard days method, and emergency contraception(4,30) during the 2019 EMDHS survey period while woman who had been utilizing traditional, folkloric or no method was considered as a “non-utilizer”. 

Comment:

In Results, please merge table 1 and 2 to avoid clutter, though they represent individual and community level results, dividing it into two sections with a heading using a single row with heading. 

Response:

Table 1 and Table 2 were merged 

Lines 183 and 184

Table 2: Distribution of individual and community-level factors of the married women in Ethiopia, EMDHS 2019

Individual variables Number (N) Percent (%)

Maternal age

15-19

20-24

25-29

30-34

35-39

40-44

45-49 

382

803

1170

870

810

546

402 

7.67

16.11

23.48

17.46

16.26

10.96

8.07

Educational level of mother

No education 

Primary 

Secondary 

Higher 

2604

1589

476

314 

52.26

31.89

9.55

6.30

Wealth index

Low Income

Middle Income

High Income 

2056

793

2134 

41.26

15.91

42.83

Total children ever born

1-3 children 

4-6 children 

7 and above 

2189

1462

1332 

43.93

29.34

26.73

Number of living children

No child

1-4 children

5-8 children

9 and above 

514

2965

1391

113 

10.32

59.50

27.91

2.27

Births in the last three years

No birth 

One birth

2 and above births 

2479

2152

352 

49.75

43.19

7.06

Age of respondent at first birth

Less than 18 years

18 – 24 

25 and above 

2509

1632

842 

50.35

32.75

16.90

Number of household members

1-4

5-8

9 and above 

1803

2652

528 

36.18

53.22

10.60

Number of under five children in household

None 

1-2

3 and above 

1521

3088

374 

30.52

61.97

7.51

Knowledge of any method

Knows no method

Knows only folkloric method 

Knows only traditional method

Knows modern method 

311

1

6

4665 

6.24

0.02

0.12

93.62

Religion 

Muslim 

Orthodox 

Protestant 

Catholic 

Other 

2,105

1,806

983

38

51 

42.24

36.24

19.73

0.76

1.02

Region 

Tigray 

Afar 

Amhara 

Oromia 

Somali 

Benishangul 

SNNPR

Gambela 

Harari 

Addis Ababa

Dire Dawa 

391

407

584

642

353

467

629

406

378

331

395 

7.85

8.17

11.72

12.88

7.08

9.37

12.62

8.15

7.59

6.64

7.93

Type of place of residence 

Urban 

Rural 

1365

3618 

27.39

72.61

Comment:

Line 185: 'did not utilize'. 

Response:

Corrected 

Lines 188-190

Accordingly, a large number of women did not utilize any contraceptive method while the least commonly utilized method was folkloric method (Fig. 1). 

Comment:

Statement in lines 279-283 completely contradict itself.

Response: 

In these statements we have tried to show the effect of religion on utilization of modern contraceptive methods. According to the findings of both studies, the odds of modern contraceptive methods utilization was lower among women who have been following Muslim religion than those women who have been following non-Muslim or other religions. Unfortunately, I cannot able to see a statement that contradict itself or else I couldn’t understand your question. 

Regarding statistical analysis:

First we have excluded those women who were unmarried or not in union during the survey period (keep if v502==1) i.e. code 1 is currently in union or living with a man. This command deleted 3,145 observations. 

Secondly, all pregnant women during the survey period were also excluded (keep if v213==0) where 0 is a code for no or unsure pregnancy. Similarly, this command deleted 759 observations. Finally 4,983 observations remained for further analysis from a total of 8,885 observations of the 2019 EMDHS. 

After identifying significant factors (p-value of <0.2) in bivariate analysis, we used a command “melogit” to run multilevel mixed effect logistic regression analysis. 

That is all what I have. Thanks!

---

## [Decision Letter · Decision Letter 1]

26 Dec 2021

Four out of ten married women utilized modern contraceptive method in Ethiopia: A multilevel analysis using the Ethiopia mini demographic and health survey 2019

PONE-D-21-20183R1

Dear Dr. Shagaro,

We’re pleased to inform you that your manuscript has been judged scientifically suitable for publication and will be formally accepted for publication once it meets all outstanding technical requirements.

Kind regards,

Faisal Abbas, PhD

Academic Editor

PLOS ONE

Additional Editor Comments (optional):

Accept.

Reviewers' comments:

Reviewer's Responses to Questions

**Comments to the Author**

1. If the authors have adequately addressed your comments raised in a previous round of review and you feel that this manuscript is now acceptable for publication, you may indicate that here to bypass the “Comments to the Author” section, enter your conflict of interest statement in the “Confidential to Editor” section, and submit your "Accept" recommendation.

Reviewer #1: All comments have been addressed

Reviewer #2: All comments have been addressed

2. Is the manuscript technically sound, and do the data support the conclusions?

Reviewer #1: Yes

Reviewer #2: Yes

3. Has the statistical analysis been performed appropriately and rigorously? 

Reviewer #1: Yes

Reviewer #2: Yes

4. Have the authors made all data underlying the findings in their manuscript fully available?

Reviewer #1: Yes

Reviewer #2: Yes

5. Is the manuscript presented in an intelligible fashion and written in standard English?

Reviewer #1: Yes

Reviewer #2: Yes

6. Review Comments to the Author

Reviewer #1: Thank you for addressing the feedback in the revised manuscript. Although the work does provide valuable data on contraceptive use in Ethiopia and the factors which predispose women to use contraceptives it is hoped that future work will elaborate more on ways to support behavioural changes toward family planning particularly addressing most vulnerable groups. This includes the benefits or otherwise of different contraceptive methods including Long Acting Reversible Contraception (LARC) preferences and policies to support contraception with vulnerable populations.

Reviewer #2: The changes have been incorporated and the queries have been satisfactorily responded to. all required questions have been answered and that all responses meet formatting specifications.

7. PLOS authors have the option to publish the peer review history of their article (what does this mean?). If published, this will include your full peer review and any attached files.

Reviewer #1: **Yes: **Tahir Turk

Reviewer #2: **Yes: **Mohammad Tahir Khan

---

## [Editor Report · Acceptance letter]

6 Jan 2022

PONE-D-21-20183R1 

Four out of ten married women utilized modern contraceptive method in Ethiopia: A Multilevel analysis of the 2019 Ethiopia mini demographic and health survey 

Dear Dr. Shagaro:

I'm pleased to inform you that your manuscript has been deemed suitable for publication in PLOS ONE. Congratulations! Your manuscript is now with our production department. 

Kind regards, 

on behalf of

Dr. Faisal Abbas 

Academic Editor

PLOS ONE